# Anomalies in the space of coupling constants and their dynamical applications II

**Clay Córdova[1⋆], Daniel S. Freed[2], Ho Tat Lam[3] and Nathan Seiberg [1]**

**1** School of Natural Sciences, Institute for Advanced Study, Princeton NJ, USA
**2** Math Department, University of Texas at Austin, Austin, TX, USA
**3** Physics Department, Princeton University, Princeton, NJ, USA

⋆ claycordova@ias.edu

## Abstract

We extend our earlier work on anomalies in the space of coupling constants to four-dimensional gauge theories. Pure Yang-Mills theory (without matter) with a simple and simply connected gauge group has a mixed anomaly between its one-form global symmetry (associated with the center) and the periodicity of the $\theta$-parameter. This anomaly is at the root of many recently discovered properties of these theories, including their phase transitions and interfaces. These new anomalies can be used to extend this understanding to systems without discrete symmetries (such as time-reversal). We also study $SU(N)$ and $Sp(N)$ gauge theories with matter in the fundamental representation. Here we find a mixed anomaly between the flavor symmetry group and the $\theta$-periodicity. Again, this anomaly unifies distinct recently-discovered phenomena in these theories and controls phase transitions and the dynamics on interfaces.

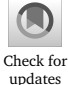

# 1 Introduction and Summary

## 1.1 Anomalies and Symmetries

't Hooft anomalies are powerful tools for analyzing strongly coupled quantum field theories (QFTs). They constrain the long-distance dynamics and control the properties of boundaries and interfaces, as well as extended excitations like strings and domain walls.

An 't Hooft anomaly can be characterized as an obstruction to coupling a system to classical background gauge fields for its global symmetries.[1] The classical background fields, denoted by $A$, have various gauge redundancies parameterized by gauge parameters $\lambda$, and the partition function $Z[A]$ is expected to be gauge invariant under these background gauge transformations $A \to A^\lambda$. When a system has an 't Hooft anomaly, one instead finds that the partition function $Z[A]$ is not gauge invariant. Rather, under gauge transformation its phase shifts by a local functional of the gauge parameters $\lambda$ and the background fields $A$:

$$Z[A^\lambda] = Z[A] \exp\left(-2\pi i \int_X \alpha(\lambda, A)\right) , \tag{1.1}$$

where $X$ is our $d$-dimensional spacetime. Of course as usual, the partition function $Z[A]$ has an ambiguity parameterized by regularization schemes. The 't Hooft anomaly is only considered to be non-trivial if it cannot be removed by a suitable choice of scheme.

In general, it is convenient to summarize 't Hooft anomalies in terms of a $(d+1)$-dimensional, classical, local action $\omega(A)$ for the gauge fields $A$ with the property that on open $(d+1)$ manifolds $Y$

$$\exp\left(2\pi i \int_Y \omega(A^\lambda) - 2\pi i \int_Y \omega(A)\right) = \exp\left(2\pi i \int_Y d\alpha(\lambda, A)\right) . \tag{1.2}$$

---

[1]These background gauge fields include standard connections for ordinary (0-form) global symmetries, which can be continuous or discrete. They also include appropriate background fields for generalized global symmetries [1]. Finally, they also include the metric and associated discrete geometric data.

Such actions $\omega(A)$ are also referred to as invertible field theories. On closed $(d+1)$-manifolds the anomaly action $\omega$ defines a gauge-invariant partition function

$$\mathcal{A}[A] = \exp\left(2\pi i \int \omega(A)\right) ,\qquad(1.3)$$

while on manifolds with boundary it reproduces the anomaly. In some condensed matter applications, the $(d+1)$-dimensional spacetime is physical. The system $X$ is on a boundary of a non-trivial SPT phase on $Y$. The 't Hooft anomaly of the boundary theory is then provided by inflow from the nontrivial bulk. This is known as anomaly inflow and was first described in [2] (see also [3]).

The anomaly action $\omega$ evolves continuously under changes in parameters or renormalization group flow, and this leads to powerful constraints on QFTs. In particular, any theory with an anomaly action $\omega$ that cannot be continuously deformed to the trivial action cannot flow at long distances to a trivially gapped theory with a unique vacuum and no long-range degrees of freedom.[2]

It is typical to discuss anomalies for global symmetries. In [4], we generalized these concepts to include the dependence on scalar coupling constants. The analysis there extends previous work on this subject in [5–8]. (For a related discussion in another context see e.g. [9].) These generalized anomalies of $d$-dimensional theories can also be summarized in terms of classical theories in $(d+1)$-dimensions but now the anomaly action $\omega$ depends non-trivially on the coupling constants viewed as background scalar fields varying over spacetime.

Important classes of examples discussed in [4] include the circle-valued $\theta$-angle in two-dimensional $U(1)$ gauge theory or in the quantum mechanics of a particle on a circle. There, the anomaly in the mass parameters of free fermions in various spacetime dimensions was also presented.[3]

As with ordinary 't Hooft anomalies, these generalized anomalies can be used to constrain the phase diagram of the theory as a function of its parameters. For instance, if the low-energy theory is nontrivial, i.e. gapless or gapped and topological, it should have the same anomaly. And if it is gapped and trivial, there must be a phase transition for some value of the parameters.[4] We can also use these generalized anomalies to learn about the worldvolume theory of defects constructed by position-dependent coupling constants. We will see examples of both applications below.

## 1.2 Anomalies in Yang-Mills Theory

In this paper, we study these generalized anomalies in four-dimensional gauge theories both with and without matter. One of our main results are formulas for the anomaly involving the $\theta$-angle of Yang-Mills theory with a general simply connected, and simple gauge group. These formulas are summarized in Table 1. We also determine the anomaly for $SU(N)$ and $Sp(N)$ gauge theory coupled to fundamental matter.

As a characteristic example of the analysis to follow, consider four-dimensional $SU(N)$ Yang-Mills theory viewed as a function of the $\theta$-angle. At long distances the theory is believed to be confining and generically has a unique ground state. However, as we will show, this

---

[2]The low-energy theory of a trivially gapped theory is a classical theory of the background fields also referred to as an invertible theory.

[3]Another simple example with an anomaly in the space of coupling constants is 4$d$ Maxwell theory viewed as a function of its $\tau$ parameter. This follows from the analysis in [8,10], though it was not presented in that language there.

[4]Discontinuities in counterterms are a common tool for deducing phase transitions. The above rephrases this logic in terms of 't Hooft anomalies. This idea has been fruitfully applied in the study of 3$d$ dualities by tracking the total discontinuity in various Chern-Simons levels for background fields as a parameter is varied [11–22].

Table 1: The anomaly involving the $\theta$-angle in Yang-Mills theory with simply connected gauge group $G$. These QFTs have a one-form symmetry, which is the center of the gauge group, $Z(G)$. (In particular, the omitted groups $G_2, F_4, E_8$ have a trivial center and hence their corresponding Yang-Mills theories do not have a one-form symmetry.) The anomaly $\omega$ depends on the background gauge field $B$ for these one-form symmetries. ($\mathcal{P}(B)$ is the Pontryagin square operation defined in footnote 7.)

| Gauge group $G$ | One-Form Sym. $(Z(G))$ | Anomaly $\omega$ |
|---|---|---|
| $SU(N)$ | $\mathbb{Z}_N$ | $\frac{N-1}{2N}\int d\theta \,\mathcal{P}(B)$ |
| $Sp(N)$ | $\mathbb{Z}_2$ | $\frac{N}{4}\int d\theta \,\mathcal{P}(B)$ |
| $E_6$ | $\mathbb{Z}_3$ | $\frac{2}{3}\int d\theta \,\mathcal{P}(B)$ |
| $E_7$ | $\mathbb{Z}_2$ | $\frac{3}{4}\int d\theta \,\mathcal{P}(B)$ |
| $Spin(N)$, $N$ odd | $\mathbb{Z}_2$ | $\frac{1}{2}\int d\theta \,\mathcal{P}(B)$ |
| $Spin(N)$, $N = 2 \bmod 4$ | $\mathbb{Z}_4$ | $\frac{N}{16}\int d\theta \,\mathcal{P}(B)$ |
| $Spin(N)$, $N = 0 \bmod 4$ | $\mathbb{Z}_2 \times \mathbb{Z}_2$ | $\frac{N}{16}\int d\theta \,\mathcal{P}(B_L + B_R) + \frac{1}{2}\int d\theta \, B_L \cup B_R$ |

conclusion cannot persist for all values of $\theta$. There should be at least one phase transition at some $\theta_* \in [0, 2\pi)$. It is commonly assumed that this phase transition takes place at $\theta_* = \pi$ and it was argued in some cases [23] that this transition follows from a mixed anomaly between the time reversal symmetry of the system and the global one-form symmetry. Here we argue for this transition without using the time reversal symmetry and therefore our results apply even when the system is deformed and that symmetry is explicitly broken.

To further expand on this example, we will proceed by carefully considering the periodicity of $\theta$. Placing the theory on $\mathbb{R}^4$ in a topologically trivial configurations of background fields, i.e. all those necessary to consider all correlation functions of local operators in flat space, the parameter $\theta$ has periodicity $2\pi$. However, when we couple to topologically non-trivial background fields the $2\pi$-periodicity is violated. Specifically, the $SU(N)$ Yang-Mills theory has a one-form symmetry $\mathbb{Z}_N^{(1)}$ measuring the transformation properties of Wilson lines under the center of $SU(N)$ [1].[5] This symmetry is intimately connected with confinement: in a deconfined phase it is spontaneously broken, in a confined phase it is preserved [1]. This one-form global symmetry means that the theory can be studied in topologically non-trivial 't Hooft twisted boundary conditions [24]. Equivalently in a somewhat more sophisticated language, the theory can be coupled to $\mathbb{Z}_N$-valued two-form gauge field $B$. In the presence of a background $B$-field the $SU(N)$ instanton number can fractionalize and the periodicity of $\theta$ is enlarged to $2\pi N$ (or $4\pi N$ for $N$ even on a non-spin manifold).

This violation of the expected periodicity of $\theta$ in the presence of background fields is conceptually very similar to the general paradigm of anomalies described above. Specifically, we find that the $2\pi$ periodicity of $\theta$ can be restored by coupling the theory to a five-dimensional classical field theory that depends on $\theta$. Its Lagrangian is

$$\omega = \frac{N-1}{2N}\frac{d\theta}{2\pi} \cup \mathcal{P}(B) \,, \tag{1.4}$$

where $\mathcal{P}(B)$ is the Pontryagin square (more details are given in section 2 below). In particular, this non-trivial anomaly must be matched under renormalization group flow now applied to

---

[5]For symmetries, we sometimes use the notation $\mathcal{G}^{(p)}$ to indicate that $\mathcal{G}$ is a $p$-form symmetry group.

the family of theories labelled by $\theta \in [0, 2\pi)$. A trivially gapped theory for all $\theta$ does not match the anomaly and hence it is excluded.

We can alternatively describe the anomaly as follows. The theories at $\theta = 0$ and $\theta = 2\pi$ differ in their coupling to background $B$ fields by a classical function of $B$ (a counterterm) $2\pi \frac{N-1}{2N} \int \mathcal{P}(B)$ [1, 23, 25, 26]. However, since the coefficient of the counterterm $2\pi \frac{1}{2N} \int \mathcal{P}(B)$ must be quantized, this difference cannot be removed by making its coefficient $\theta$-dependent in a smooth fashion. This means that at some $\theta_* \in [0, 2\pi)$ the vacuum must become non-trivial so that the counterterm can jump discontinuously. For instance in the case of $SU(N)$ Yang-Mills it is believed that at $\theta = \pi$ the theory has two degenerate vacua. See [27–33] for early references about the dynamics at $\theta = \pi$ in two and four dimensions.

We can also use this example to clarify the precise relationship of our results to previous analyses of time-reversal anomalies in Yang-Mills theory discussed in [23]. $SU(N)$ Yang-Mills theory is time-reversal ($\mathsf{T}$) invariant at the two values $\theta = 0$ and $\theta = \pi$. For even $N$ when $\theta = \pi$, there is a mixed anomaly between $\mathsf{T}$ and the one-form $\mathbb{Z}_N^{(1)}$ global symmetry, and hence the long-distance behavior at $\theta = \pi$ cannot be trivial in agreement with the general discussion above. For odd $N$ the situation is more subtle. In this case there is no $\mathsf{T}$ anomaly for $\theta$ either 0 or $\pi$, but it is not possible to write continuous counterterms as a function of $\theta$ that preserves $\mathsf{T}$ in the presence of background $B$ fields at both $\theta = 0, \pi$ [23]. (This situation was referred to in [34, 35] as "a global inconsistency.") Again this implies that there must be a phase transition at some value of $\theta$ in agreement with our conclusion above.

Thus, while our conclusions agree with previous results, they also generalize them in new directions. Indeed, the focus of the previous analysis is on subtle aspects of $\mathsf{T}$ symmetry, while in the anomaly in the space of parameters (1.4) $\mathsf{T}$ plays no role. As emphasized above, this means that the anomaly in the space of parameters, and consequently our resulting dynamical conclusions, persists under $\mathsf{T}$-violating deformations.

## 1.3 Worldvolume Anomalies on Interfaces in Yang-Mills

As discussed in detail in [4], another application of anomalies in the space of coupling constants is to determine the worldvolume anomaly on defects defined by varying parameters. This is because these generalized 't Hooft anomalies can also be viewed as an obstruction to promoting the coupling constants to be position dependent.

In this paper, we will consider interfaces defined by varying the $\theta$-angle.[6] We let $\theta$ depend on a single spacetime coordinate $x$ and wind around the circle as $x$ varies from $-\infty$ to $+\infty$. If the parameter variation is smooth, i.e. it takes place over a distance scale longer than the UV cutoff, then the resulting interface dynamics is completely determined by the UV theory. Such interfaces have been widely studied recently in 4$d$ QCD and related applications to 3$d$ dualities [26, 36–38].

In each of the 4$d$ systems where we determine the anomaly, we can apply the formula to determine the worldvolume anomaly on the interface by simply evaluating it on the appropriate configuration with varying $\theta$. To illustrate this essential point, let us return again to the example of four-dimensional $SU(N)$ Yang-Mills theory. The same anomaly action (1.4) introduced to restore the $2\pi$ periodicity of $\theta$ in the presence of a background $B$ field can be used to compute the worldvolume anomaly of an interface where $\theta$ varies smoothly from 0 to $2\pi$

$$\omega_{\text{interface}} = \frac{N-1}{2N} \mathcal{P}(B) \, . \tag{1.5}$$

This correctly reproduces the anomaly in the one-form global symmetry on the interface deduced in [1,26,37]. In particular, the world-volume dynamics on the defect must be non-trivial.

---

[6]Interfaces should be distinguished from domain walls. The latter are dynamical excitations and their position is not fixed. By contrast, interfaces are pinned by the external variation of the parameters.

## 1.4   Examples and Summary

In section 2 we study four-dimensional Yang-Mills theories with a simply-connected gauge group $G$ and determine the anomaly (reproduced in Table 1). Using the logic discussed above, we also determine the worldvolume anomaly of interfaces interpolating between $\theta$ and $\theta + 2\pi k$ for some integer $k$. The anomaly constraints on the interfaces can be satisfied by the corresponding Chern-Simons theory with level $k$, $G_k$. However, as emphasized in [37], there are other options for the theory on the interface, all with the same anomaly. These generalized anomalies are invariant under deformations that preserve the center one-form symmetries. For instance, by adding appropriate adjoint Higgs field we show that the long distance theory can flow to a conformal field theory or a TQFT. These long-distance theories also reproduce the same generalized anomaly.

In section 3 we expose a geometric viewpoint on the mixed anomaly between $\theta$ and the center symmetry. We explain its origin in a failure of integrality, that is, the division of an integral characteristic class by a positive integer. We also illustrate some computational techniques.

In section 4 we extend our analysis to four-dimensional $SU(N)$ and $Sp(N)$ gauge theories with massive fundamental fermion matter. We will show that, depending on the number of fundamental flavors $N_f$, these theories have a mixed anomaly involving $\theta$ and the appropriate zero-form global symmetries. A new ingredient is that there can be nontrivial counterterms with smoothly varying coefficients which can potentially cancel the putative anomalies. For $SU(N)$ we find that the anomaly is valued in $\mathbb{Z}_L$ with $L = \gcd(N, N_f)$. In particular the anomaly is non-trivial if and only if $\gcd(N, N_f) > 1$. For $Sp(N)$ we find a $\mathbb{Z}_2$ anomaly, which is non-trivial if and only if $N$ is odd and $N_f$ is even. As in the pure gauge theory, the discussion of these anomalies extends previous analyses that rely on time-reversal symmetry.

We also use these generalized anomalies to constrain interfaces. These anomaly constraints can be saturated by an appropriate Chern-Simons matter theory. Our analysis extends the recent results about interfaces in 4d QCD in [26] and explains the relation between them and the earlier results about anomalies in 3d Chern-Simons-matter theory in [15].

In Appendix A, we discuss a different presentation of anomalies using additional higher-form gauge fields and apply it to the anomaly involving $\theta$-angles in four-dimensional gauge theories.

# 2   4d Yang-Mills Theory I

In this section we compute the anomaly of 4d Yang-Mills theories with simply connected and simple gauge groups. We use our results to compute the anomaly on interfaces with spatially varying $\theta$.

## 2.1   $SU(N)$ Yang-Mills Theory

We begin with the 4d $SU(N)$ gauge theory with the Euclidean action

$$S = -\frac{1}{4g^2} \int \mathrm{Tr}(f \wedge *f) - \frac{i\theta}{8\pi^2} \int \mathrm{Tr}(f \wedge f). \qquad (2.1)$$

Since the instanton number is quantized, the transformation $\theta \to \theta + 2\pi$ does not affect correlation functions of local operators at separated points, but it may affect more subtle observables such as contact terms involving surface operators.

The theory has a $\mathbb{Z}_N$ one-form symmetry that acts by shifting the connection by a $\mathbb{Z}_N$ connection [1]. We can turn on a background $\mathbb{Z}_N$ two-form gauge field $B \in H^2(X, \mathbb{Z}_N)$ for

this one-form symmetry. In the presence of this background gauge field, the $SU(N)$ bundle is twisted into a $PSU(N)$ bundle with fixed second Stiefel-Whitney class $w_2(a) = B$ [1,25].

The instanton number of a $PSU(N)$ bundle can be fractional. Therefore with a nontrivial background $B$ the partition function at $\theta + 2\pi$ and $\theta$ can be different [1,23,25]

$$\frac{Z[\theta + 2\pi, B]}{Z[\theta, B]} = \exp\left(2\pi i \frac{N-1}{2N} \int \mathcal{P}(B)\right), \tag{2.2}$$

where $\mathcal{P}$ is the Pontryagin square operation.[7] Thus, the theories at $\theta$ and $\theta + 2\pi$ differ by an invertible field theory, which can be detected by the contact terms of the two-dimensional symmetry operators of the $\mathbb{Z}_N$ one-form symmetry [25].

We can also add to the theory a counterterm

$$\mathcal{S} \supset -2\pi i \int_X \frac{p}{2N} \mathcal{P}(B). \tag{2.3}$$

The coefficient $p$ is an integer modulo $2N$ for even $N$ and it is an even integer modulo $2N$ for odd $N$. The difference between $\theta$ and $\theta + 2\pi$ in (2.2) can be summarized into the following identification [1,23,25,26]

$$(\theta, p) \sim (\theta + 2\pi, p + 1 - N). \tag{2.4}$$

This means that $\theta$ has an extended periodicity of $4\pi N$ for even $N$ and $2\pi N$ for odd $N$.

As discussed in detail in [4], the above phenomenon can be interpreted as a mixed anomaly between the $2\pi$-periodicity of $\theta$ and the $\mathbb{Z}_N$ one-form symmetry. The corresponding anomaly action is

$$\mathcal{A}(\theta, B) = \exp\left(2\pi i \frac{N-1}{2N} \int \frac{d\theta}{2\pi} \mathcal{P}(B)\right). \tag{2.5}$$

This anomaly implies that the long distance theory cannot be trivially gapped everywhere between $\theta$ and $\theta + 2\pi$.

We can further constrain the long distance theory using the time-reversal symmetry $\mathsf{T}$ at $\theta = 0, \pi$ following [23]. In a nontrivial background $B$, the time-reversal symmetry transforms the partition function as

$$Z[\pi, B] \to Z[\pi, B] \exp\left(2\pi i \frac{1 - N - 2p}{2N} \int \mathcal{P}(B)\right). \tag{2.6}$$

A $\mathsf{T}$ anomaly occurs if there is no value of $p$ such that the partition function is exactly invariant i.e. only if

$$1 - N - 2p = 0 \bmod 2N \tag{2.7}$$

has no integral solutions $p$. This is the case for even $N$, and hence for even $N$ there must be non-trivial long distance physics at $\theta = \pi$ [23].[8] For odd $N$, we can solve the equation above with $p$ even by taking

$$p = \begin{cases} \frac{1-N}{2} & N = 1 \mod 4, \\ \frac{1+N}{2} & N = 3 \mod 4. \end{cases} \tag{2.8}$$

Therefore, for odd $N$ there is no $\mathsf{T}$ anomaly. However, for odd $N$ the counterterm that preserves time-reversal symmetry at $\theta = 0$ has coefficient $p = 0 \bmod 2N$ and it is different from the one at $\theta = \pi$. This means that there is no continuous counterterm that preserves $\mathsf{T}$ at both $\theta = 0$ and $\pi$ [23]. (Such reasoning was named a "global inconsistency" in [34,35].) This again implies non-trivial long distance physics for at least one value of $\theta$.

---

[7]For odd $N$, $\mathcal{P}(B) = B \cup B \in H^4(X, \mathbb{Z}_N)$. For even $N$, $\mathcal{P}(B) \in H^4(X, \mathbb{Z}_{2N})$ and reduces to $B \cup B$ modulo $N$.

[8]In this case the anomaly $\omega$ is $\frac{1}{2}\tilde{w}_1 \cup \mathcal{P}(B)$, where $\tilde{w}_1 \in H^1(Y, \tilde{\mathbb{Z}})$ denotes the natural integral uplift of the Stiefel-Whitney class $w_1$ with twisted integral coefficients. For further recent discussion of time-reversal anomalies in Yang-Mills theories see also [39,40].

Table 2: Summary of anomalies and existence of continuous counterterms in various 4$d$ theories. The superscripts of the symmetries label the $q$'s of $q$-form symmetries.

| theory | without T | with T at $\theta = 0, \pi$ | |
|---|---|---|---|
| symmetry $\mathcal{G}$ | $\theta$-$\mathcal{G}$ anomaly | T-$\mathcal{G}$ anomaly at $\theta = \pi$ | no smooth counterterms |
| $SU(N)$ gauge theory $\mathcal{G} = \mathbb{Z}_N^{(1)}$ | ✔ | even $N$ ✔ <br> odd $N$ ✘ | even $N$ ✔ <br> odd $N$ ✔ |
| with adjoint scalars $\mathcal{G} = \mathbb{Z}_N^{(1)}$ | ✔ | no T symmetry in general | no T symmetry in general |

These results agree with the standard lore about Yang-Mills theory. For all values of $\theta$ the theory is confined (so the $\mathbb{Z}_N$ one-form symmetry is unbroken [1]) and gapped. For $\theta \neq \pi$ there is a unique vacuum. While at $\theta = \pi$ the T symmetry is spontaneously broken leading to two degenerate vacua and hence a first order phase transition.

We can also use the anomaly (2.5) to constrain the worldvolume of interfaces where $\theta$ varies. Consider a smooth interface between $\theta$ and $\theta + 2\pi k$. Assuming that the $SU(N)$ gauge theory is gapped at long distances, the interface supports an isolated 3$d$ quantum field theory. The anomaly (2.5) implies that the interface theory has an anomaly associated to the $\mathbb{Z}_N$ one-form symmetry described by [1, 26, 37]

$$\mathcal{A}(B) = \exp\left( 2\pi i k \int \frac{N-1}{2N} \mathcal{P}(B) \right).$$ (2.9)

The anomaly can be saturated for instance, by an $SU(N)_k$ Chern-Simons theory or a $(\mathbb{Z}_N)_{-N(N-1)k}$ discrete gauge theory [37].

## 2.2 Adding Adjoint Higgs Fields

The mixed anomaly between the $2\pi$-periodicity of $\theta$ and the $\mathbb{Z}_N$ one-form symmetry is robust under deformations that preserve the one-form symmetry. Note that such deformations generally break the time-reversal symmetry at $\theta = 0, \pi$. Below, we present two examples with different infrared behaviors that also saturate the anomaly by adding charged scalars in the adjoint representation.

As in [37], we can add one adjoint scalar to Higgs the $SU(N)$ gauge field to its Cartan torus $U(1)^{N-1}$ with gauge fields $a_J$. The $U(1)$ gauge fields are embedded in the $SU(N)$ gauge field through

$$a = \sum_{J=1}^{N-1} a_J T^J, \quad T^J = \mathrm{diag}(0, \cdots, 0, \underbrace{+1, -1}_{J\text{th entry}}, 0, \cdots, 0).$$ (2.10)

In the classical approximation, the low energy $U(1)^{N-1}$ gauge theory is described by the Euclidean action

$$S = -\frac{1}{4g^2} \int \sum_{I,J=1}^{N-1} K_{IJ} da_I \wedge *da_J - \frac{i\theta}{8\pi^2} \int \sum_{I,J=1}^{N-1} K_{IJ} da_I \wedge da_J,$$ (2.11)

where $K$ is the Cartan matrix of $SU(N)$. Small higher order quantum corrections renormalize the gauge coupling $g$ and $\theta$, but do not affect our conclusions.

The low-energy theory exhibits a spontaneously broken $U(1)^{N-1} \times U(1)^{N-1}$ one-form global symmetry. Most of it is accidental. The exact one-form symmetry is the symmetry

in the UV, which is $\mathbb{Z}_N$. It acts on the infrared fields as

$$a_J \rightarrow a_J + \frac{2\pi J}{N}\epsilon\,, \tag{2.12}$$

where $\epsilon$ is a flat connection with $\mathbb{Z}_N$ holonomies. Activating the background gauge field $B \in H^2(X, \mathbb{Z}_N)$ for the one-form symmetry modifies the Euclidean action by replacing $da_I$ with $da_I - \frac{2\pi I}{N}B$. This means that when $\theta$ is shifted by $2\pi$, the partition function of the infrared theory transforms as

$$Z[\theta + 2\pi, B] = Z[\theta, B]\exp\left(2\pi i \frac{N-1}{2N}\int \mathcal{P}(B)\right)\,, \tag{2.13}$$

which agrees with the anomaly in the ultraviolet theory.

Note that this gapless $U(1)^{N-1}$ gauge theory reproduces the anomaly (2.5), without a phase transition.

Following [37], we can also add more adjoint scalars to Higgs the theory to a $\mathbb{Z}_N$ gauge theory. The $\mathbb{Z}_N$ gauge field $c$ is embedded in the $SU(N)$ gauge field through (we work in continuous notation, i.e. $c$ is a flat $U(1)$ gauge field with holonomies in $\mathbb{Z}_N$)

$$a = cT, \quad T = \text{diag}(1, \cdots, 1, -(N-1))\,. \tag{2.14}$$

The infrared theory is a topological field theory with Euclidean action

$$S = \frac{iN}{2\pi}\int b \wedge dc - N(N-1)\frac{i\theta}{8\pi^2}\int dc \wedge dc\,, \tag{2.15}$$

where $b$ is a dynamical $U(1)$ two-form gauge field and $c$ is a dynamical $U(1)$ one-form gauge field. $b$ acts as a Lagrange multiplier constraining $c$ to be a $\mathbb{Z}_N$ gauge field. The equation of motion of $b$ constrains $c$ to be a $\mathbb{Z}_N$ gauge field that satisfies $Nc = d\phi$. The original $\mathbb{Z}_N$ one-form symmetry is spontaneously broken in the infrared. If we activate the background gauge field $B$ for the $\mathbb{Z}_N$ one-form symmetry. The Euclidean action becomes

$$S = \frac{iN}{2\pi}\int b \wedge \left(dc - \frac{2\pi}{N}B\right) - N(N-1)\frac{i\theta}{8\pi^2}\int \left(dc - \frac{2\pi}{N}B\right) \wedge \left(dc - \frac{2\pi}{N}B\right)\,. \tag{2.16}$$

As the coupling constant $\theta$ shifts by $2\pi$, the partition function of the infrared theory transforms anomalously and agrees with the anomaly in ultraviolet theory. As in the gapless $U(1)^{N-1}$ theory discussed above, the anomaly is saturated in the IR without a phase transition.

We can also simplify the above $\mathbb{Z}_N$ gauge theory by shifting $b \rightarrow b + \frac{N-1}{4\pi}\theta dc$. The Euclidean action then becomes that of a standard $\mathbb{Z}_N$ gauge theory [41, 42]

$$S = \frac{iN}{2\pi}\int b \wedge dc\,. \tag{2.17}$$

The dependance on $\theta$ now appears in the coupling of these fields to the background $B$ and the partition function again transforms anomalously when $\theta \rightarrow \theta + 2\pi$ in agreement with (2.2) and the ultraviolet anomaly (2.5). Again, this is achieved in the IR without a phase transition.

## 2.3 Other Gauge Groups

We now discuss similar mixed anomalies involving the $2\pi$-periodicity of $\theta$ and the center one-form symmetries in 4$d$ Yang-Mills theories with other simply-connected gauge groups $G$. These anomalies constrain the long distance physics of these theories as well as smooth interfaces separating two regions with different $\theta$'s. As we will see, unlike the case of $SU(N)$, which we

studied above, typically $2\pi$ shifts of $\theta$ do not allow us to scan all the possible values of the coefficient $p$ of the $\mathcal{P}(B)$ counterterm.

The one-form global symmetry of any of these simply connected groups $G$ is its center $Z(G)$. We couple it to a two-form gauge field $B$. This twists the gauge bundles to $G/Z(G)$ bundles with second Stiefel-Whitney (SW) class $w_2 = B$. These bundles support fractional instantons. Following [43], we will determine the relation between the fractional instantons and the background gauge fields $B$ by evaluating the instanton number on a specific $G/Z(G)$ bundle. We will take it to be of a tensor product of various $SU(n)/\mathbb{Z}_n$ bundles, for which we already know the answer, and untwisted bundles of simply connected groups. We will generalize the discussion in [43] to non-spin manifolds.

We will discuss $Sp(N)$, $Spin(N)$, $E_6$ and $E_7$ gauge groups. The other simple Lie groups $G_2$, $F_2$, and $E_8$ have trivial center and therefore the corresponding gauge theories do not have similar anomalies.

### 2.3.1 $Sp(N)$ **Gauge Theory**

We start with a pure gauge $Sp(N)$ theory.[9] The theory has a $\mathbb{Z}_2$ one-form symmetry. We want to construct a $Sp(N)/\mathbb{Z}_2$ bundle with second SW class $B$. We do that by using the embedding $SU(2)^N \subset Sp(N)$ and then an $Sp(N)/\mathbb{Z}_2$ bundle is found by tensoring $N$ $PSU(2)$ bundles each with second SW class $B$. Then the anomaly (2.5) implies that the $Sp(N)$ gauge theory has an anomaly

$$\mathcal{A}_{Sp(N)}(\theta, B) = \mathcal{A}_{SU(2)}(\theta, B)^N = \exp\left(2\pi i \int \frac{d\theta}{2\pi} \frac{N\mathcal{P}(B)}{4}\right). \tag{2.18}$$

This means that a shift of $\theta$ by $2\pi$ shifts the coefficient $p$ of the counterterm $2\pi i p \int \frac{\mathcal{P}(B)}{4}$ by $N$. Note that for even $N$ not all the possible values of $p = 0, 1, 2, 3$ are scanned by shifts of $\theta$ by $2\pi$. The anomaly becomes trivial when $N = 0 \bmod 4$ (on spin manifolds it is trivial when $N = 0 \bmod 2$).

### 2.3.2 $E_6$ **Gauge Theory**

The theory has a $\mathbb{Z}_3$ one-form symmetry. Here we use the embedding $SU(3)^3 \subset E_6$. We can construct a $E_6/\mathbb{Z}_3$ bundle with second SW class $B$ by tensoring an $SU(3)$ bundle, a $PSU(3)$ bundle with second SW class $B$, and a $PSU(3)$ bundle with second SW class $-B$. Then the anomaly (2.5) implies that the $E_6$ gauge theory has an anomaly

$$\mathcal{A}_{E_6}(\theta, B) = \mathcal{A}_{SU(3)}(\theta, B)^2 = \exp\left(2\pi i \int \frac{d\theta}{2\pi} \frac{2\mathcal{P}(B)}{3}\right). \tag{2.19}$$

The anomaly is nontrivial and all possible values of $p$ in the counterterm are scanned by shifts of $\theta$ by $2\pi$.

### 2.3.3 $E_7$ **Gauge Theory**

The theory has a $\mathbb{Z}_2$ one-form symmetry. Here we use the embedding $SU(4) \times SU(4) \times SU(2) \subset E_7$. We can construct a $E_7/\mathbb{Z}_2$ bundles with second SW class $B$ by tensoring an $SU(4)$ bundle, a $PSU(2)$ bundle with second Stifel-Whitney class $B$, and an $SU(4)/\mathbb{Z}_2$ bundle with second SW class $B$ (which can be thought of as $SU(4)/\mathbb{Z}_4$ bundle with second SW class $2\widetilde{B}$ where the tilde denotes a lifting to a $\mathbb{Z}_4$ cochain and $2\widetilde{B}$ is independent of the lift). Then the anomaly (2.5) implies that the $E_7$ gauge theory has an anomaly

$$\mathcal{A}_{E_7}(\theta, B) = \mathcal{A}_{SU(2)}(\theta, B)\mathcal{A}_{SU(4)}(\theta, 2\widetilde{B}) = \exp\left(2\pi i \int \frac{d\theta}{2\pi} \frac{3\mathcal{P}(B)}{4}\right). \tag{2.20}$$

---

[9]We use the notation $Sp(N) = USp(2N)$. Specifically $Sp(1) = SU(2)$ and $Sp(2) = Spin(5)$.

Again, the anomaly is nontrivial and all possible values of $p$ in the counterterm are scanned.

### 2.3.4 $Spin(N) = Spin(2n + 1)$ Gauge Theory

For $N = 3$ this is the same as $SU(2)$, which was discussed above. So let us consider $N \geqslant 5$.

The theory has a $\mathbb{Z}_2$ one-form symmetry. Here we use the embedding $SU(2) \times SU(2) \times Spin(N - 4) \subset Spin(N)$ (where the last factor is missing for $N = 5$). We can construct a $Spin(N)/\mathbb{Z}_2$ bundle with second SW class $B$ by tensoring two $PSU(2)$ bundles each with second SW class $B$ and a $Spin(N - 4)$ bundle. Then the anomaly (2.5) implies that the $Spin(N) = Spin(2n + 1)$ gauge theory has an anomaly

$$\mathcal{A}_{Spin(N)}(\theta, B) = \mathcal{A}_{SU(2)}(\theta, B)^2 = \exp\left(2\pi i \int \frac{d\theta}{2\pi} \frac{\mathcal{P}(B)}{2}\right) \quad \text{for} \quad N = 1 \bmod 2. \quad (2.21)$$

The anomaly is always nontrivial (but it is trivial on spin manifolds). A shift of $\theta$ by $2\pi$ shifts $p$ by 2 and hence not all values of $p$ are scanned by such shifts.

### 2.3.5 $Spin(N) = Spin(4n + 2)$ Gauge Theory

For $N = 6$ this is the same as $SU(4)$ which was discussed above. So we will discuss here $N \geqslant 10$.

The theory has a $\mathbb{Z}_4$ one-form symmetry. We will use the embedding $Spin(6) \times Spin(4)^{n-1} \subset Spin(N)$. We can construct a $Spin(N)/\mathbb{Z}_4$ bundle with second SW class $B$ by tensoring $(n-1)$ $SU(2)$ bundles, $(n-1)$ $PSU(2)$ bundles each with second SW class $B$ mod 2 and a $PSU(4)$ bundle with second SW class $B$. Then the anomaly (2.5) implies that the $Spin(N) = Spin(4n + 2)$ gauge theory has an anomaly[10]

$$\begin{aligned}
\mathcal{A}_{Spin(N)}(\theta, B) &= \mathcal{A}_{SU(4)}(\theta, B)\mathcal{A}_{SU(2)}(\theta, B)^{n-1} \\
&= \exp\left(2\pi i \int \frac{d\theta}{2\pi} \frac{N\mathcal{P}(B)}{16}\right) \quad \text{for} \quad N = 2 \bmod 4.
\end{aligned} \quad (2.22)$$

The anomaly is always nontrivial (even on spin manifolds). A shift of $\theta$ by $2\pi$ shifts $p$ by $\frac{N}{2}$ and hence all values of $p$ are scanned by such shifts.

If $B = 2\widehat{B}$ is even, we study $SO(N)$ bundles and the anomaly is

$$\exp\left(2\pi i \int \frac{d\theta}{2\pi} \frac{N\mathcal{P}(\widehat{B})}{4}\right) = \exp\left(2\pi i \int \frac{d\theta}{2\pi} \frac{\mathcal{P}(\widehat{B})}{2}\right). \quad (2.23)$$

This is useful, e.g. when we add dynamical matter fields in the vector representation and the one-form global symmetry is only $\mathbb{Z}_2 \subset \mathbb{Z}_4$, which is coupled to $\widehat{B}$. In that case the anomaly vanishes on spin manifolds and a shift of $\theta$ by $2\pi$ shifts the coefficient $p$ of the counterterm $2\pi i p \int \frac{\mathcal{P}(\widehat{B})}{4}$ by 2 and hence not all possible values of $p$ are scanned. This is the same conclusion as for odd $N$ (2.21).

---

[10]The instanton number of a $Spin(4n + 2)/\mathbb{Z}_4$ bundle with second SW class $B$ is $\int \frac{2n+1}{4} \frac{\mathcal{P}(B)}{2}$ mod 1. On spin manifolds $\frac{\mathcal{P}(B)}{2} \in H^4(X, \mathbb{Z}_4)$, so for $N = 4n + 2 = 2 \bmod 8$ the instanton number is $\int \frac{1}{4} \frac{\mathcal{P}(B)}{2}$ mod 1, while for $N = 4n + 2 = 6 \bmod 8$ the instanton number is $-\int \frac{1}{4} \frac{\mathcal{P}(B)}{2}$ mod 1. For $N = 6 \bmod 8$, our determination of the fractional instanton number on spin manifolds differs from [43] by a sign. However it does not affect the computation of the supersymmetric index in [43]. The discrepancy propagates to [44]. This sign change reverses the direction of the action of the modular T-transformation in Fig. 6 of [44] for $N = 6 \bmod 8$.

### 2.3.6 $Spin(N) = Spin(4n)$ **Gauge Theory**

The theory has a $\mathbb{Z}_2^{(L)} \times \mathbb{Z}_2^{(R)}$ one-form symmetry. Here we use the embedding $SU(2)^{2n} \subset Spin(N)$.

For odd $n$ we can construct a $Spin(N)/(\mathbb{Z}_2^{(L)} \times \mathbb{Z}_2^{(R)})$ bundle with second SW class $B_L$ and $B_R$ by tensoring $n$ $PSU(2)$ bundles with second SW class $B_L$ and $n$ $PSU(2)$ bundles with second SW class $B_R$. Then the anomaly (2.5) implies that the $Spin(N) = Spin(4n)$ gauge theory for odd $n$ has an anomaly

$$
\begin{aligned}
\mathcal{A}_{Spin(N)}(\theta, B_L, B_R) &= \mathcal{A}_{SU(2)}(\theta, B_L)^n \mathcal{A}_{SU(2)}(\theta, B_R)^n \\
&= \exp\left( 2\pi i \int \frac{d\theta}{2\pi} \frac{N\left(\mathcal{P}(B_L) + \mathcal{P}(B_R)\right)}{16} \right) \quad \text{for} \quad N = 4 \bmod 8 \,.
\end{aligned}
\tag{2.24}
$$

For even $n$ we can construct the $Spin(N)/(\mathbb{Z}_2^{(L)} \times \mathbb{Z}_2^{(R)})$ bundle by tensoring an $SU(2)$ bundle, a $PSU(2)$ bundle with second SW class $B_L + B_R$, $n-1$ $PSU(2)$ bundles with second SW class $B_L$, and $n-1$ $PSU(2)$ bundles with second SW class $B_R$. Then the anomaly (2.5) implies that the $Spin(N) = Spin(4n)$ gauge theory for even $n$ has an anomaly

$$
\begin{aligned}
\mathcal{A}_{Spin(N)}(\theta, B_L, B_R) &= \mathcal{A}_{SU(2)}(\theta, B_L)^{n-1} \mathcal{A}_{SU(2)}(\theta, B_R)^{n-1} \mathcal{A}_{SU(2)}(\theta, B_L + B_R) \\
&= \exp\left( 2\pi i \int \frac{d\theta}{2\pi} \left( \frac{N\left(\mathcal{P}(B_L) + \mathcal{P}(B_R)\right)}{16} + \frac{B_L \cup B_R}{2} \right) \right) \quad \text{for} \quad N = 0 \bmod 8 \,.
\end{aligned}
\tag{2.25}
$$

The two cases can be summarized as

$$
\begin{aligned}
\mathcal{A}_{Spin(N)}&(\theta, B_L, B_R) \\
&= \exp\left( 2\pi i \int \frac{d\theta}{2\pi} \left( \frac{N\mathcal{P}(B_L + B_R)}{16} + \frac{B_L \cup B_R}{2} \right) \right) \quad \text{for} \quad N = 0 \bmod 4 \,.
\end{aligned}
\tag{2.26}
$$

The anomaly is always nontrivial (even on spin manifolds). A shift of $\theta$ by $2\pi$ shifts the coefficients of the counterterm $2\pi i p_L \int \frac{\mathcal{P}(B_L)}{4} + 2\pi i p_R \int \frac{\mathcal{P}(B_R)}{4} + 2\pi i p_{LR} \int \frac{B_L \cup B_R}{2}$ by $(p_L, p_R, p_{LR}) \to (p_L + \frac{N}{4}, p_R + \frac{N}{4}, p_{LR} + 1 + \frac{N}{4})$ and hence not all values of $(p_L, p_R, p_{LR})$ are scanned by such shifts.

As above, if we limit ourselves to $SO(N)$ bundles (as is the case, e.g. when we add dynamical matter fields in a vector representation), we study backgrounds with $B_L = B_R = \widehat{B}$. Then, the anomaly is

$$
\exp\left( 2\pi i \int \frac{d\theta}{2\pi} \frac{\mathcal{P}(\widehat{B})}{2} \right)
\tag{2.27}
$$

and it vanishes on spin manifolds. A shift of $\theta$ by $2\pi$ shifts the value the coefficient $p$ in $2\pi i p \int \frac{\mathcal{P}(\widehat{B})}{4}$ by 2 and again, not all values of $p$ are scanned. This is the same conclusion as for odd $N$ (2.21) and for $N = 2 \bmod 4$ (2.23).

### 2.3.7 A Check Using $3d$ TQFT or $2d$ RCFT Considerations

One way of viewing our anomaly is as the anomaly in a one-form global symmetry in the theory along interfaces separating $\theta$ and $\theta + 2\pi k$. General considerations show that in this case of a simple and semi-simple gauge group this anomaly can always be saturated by a Chern-Simons theory with gauge group $G$ and level $k$. In 3d TQFTs, the anomaly of one-form symmetries can be determined by the spins of the lines generating the symmetry [37]. The one-form symmetries and the spins of the generating lines of various Chern-Simons theories with level 1 are summarized in Table 3. These results can be found by studying the 3d TQFT or by studying the corresponding 2d Kac-Moody algebra.

Table 3: Summary of the center one-form symmetries and the spins of the generating lines in various $3d$ Chern-Simons theories. (See the discussion in [37].) Here the gauge group is $G$ and the level is $k = 1$, i.e. this is the TQFT $G_1$. When the center is $\mathbb{Z}_\ell$, the symmetry lines are $\{1, a, \cdots, a^{\ell-1}\}$ generated by the generating line $a$. When the center is $\mathbb{Z}_2 \times \mathbb{Z}_2$, the symmetry lines are $\{1, a, b, ab\}$. The spins of these lines are denoted by $h_a$, $h_b$ and $h_{ab}$. In the case $Spin(N)$ with $N = 2$ mod 4 we also included the spin of the line $a^2$, which is used in the text. Note that in the context of $3d$ TQFT only the spin $h$ modulo one is meaningful. The values in the table are those of the conformal dimensions of the corresponding Kac-Moody representation.

| Gauge group $G$ | Center $Z(G)$ | Spins | Anomaly |
|---|---|---|---|
| $SU(N)$ | $\mathbb{Z}_N$ | $h_a = \frac{N-1}{2N}$ | $\frac{N-1}{2N} \int d\theta\, \mathcal{P}(B)$ |
| $Sp(N)$ | $\mathbb{Z}_2$ | $h_a = \frac{N}{4}$ | $\frac{N}{4} \int d\theta\, \mathcal{P}(B)$ |
| $E_6$ | $\mathbb{Z}_3$ | $h_a = \frac{2}{3}$ | $\frac{2}{3} \int d\theta\, \mathcal{P}(B)$ |
| $E_7$ | $\mathbb{Z}_2$ | $h_a = \frac{3}{4}$ | $\frac{3}{4} \int d\theta\, \mathcal{P}(B)$ |
| $Spin(N)$ $(N = 2n+1)$ | $\mathbb{Z}_2$ | $h_a = \frac{1}{2}$ | $\frac{1}{2} \int d\theta\, \mathcal{P}(B)$ |
| $Spin(N)$ $(N = 4n+2)$ | $\mathbb{Z}_4$ | $h_a = \frac{N}{16}$ $h_{a^2} = \frac{1}{2}$ | $\frac{N}{16} \int d\theta\, \mathcal{P}(B)$ |
| $Spin(N)$ $(N = 4n)$ | $\mathbb{Z}_2 \times \mathbb{Z}_2$ | $h_a = h_b = \frac{N}{16}$ $h_{ab} = \frac{1}{2}$ | $\frac{N}{16} \int d\theta\, \mathcal{P}(B_L + B_R)$ $+ \frac{1}{2} \int d\theta\, B_L \cup B_R$ |

We can use these to check the anomaly we determined using $4d$ instantons above. When the one-form symmetry is $\mathbb{Z}_\ell$ it is generated by a line $a$ such that $a^\ell = 1$. The coefficient of the anomaly is the spin of the line $a$, $h_a$ mod 1. Indeed, for $SU(N)$, $Sp(N)$, $E_6$, $E_7$, $Spin(2n+1)$, $Spin(4n+2)$, where the one-form symmetry is $\mathbb{Z}_\ell$ for some $\ell$, the anomaly is (2.5), (2.18), (2.19), (2.20), (2.21), (2.22) respectively, in agreement with the entries in Table 3. In the case of $Spin(4n)$, the global symmetry is $\mathbb{Z}_2 \times \mathbb{Z}_2$ and it is generated by two lines $a$ and $b$. In this case we have more kinds of anomalies. If either $B_L$ or $B_R$ vanishes, we can match the coefficient of $\mathcal{P}(B_R)$ and of $\mathcal{P}(B_L)$ in (2.26) with the spins of the lines. The coefficient of the mixed term can be checked by comparing the spin of the line $ab$ with the anomaly for $B_L = B_R$.

We can also focus on the $\mathbb{Z}_2$ subgroup of the one form symmetry for $Spin(N)$ for even $N$ that we discussed above. Its generating line, $a^2$ for $N = 4n + 2$ or $ab$ for $N = 4n$, has spin $\frac{1}{2}$ mod 1, which is the same as the $\mathbb{Z}_2$ generating line for $Spin(N)$ with odd $N$. This is consistent with the fact that the anomaly for this symmetry (2.21), (2.23), (2.27) is the same for all $N$.

## 3  $4d$ Yang-Mills Theory II

In this section we derive the anomaly in pure $4d$ gauge theory from a geometric viewpoint. We begin in §3.1 with the gauge group $SU(2)$. This allows us to explain the essential geometric ideas with minimal input from the topology of Lie groups. Then in §3.2 we offer techniques to compute the anomaly for more general Lie groups.

Quantum Yang-Mills theory has a single fluctuating field—the gauge field, or connection. The background fields are a Riemannian metric, $\mathbb{R}/2\pi\mathbb{Z}$-valued function $\theta$, and an orientation. (To account for time-reversal symmetry we eventually drop the orientation.) In the classical theory the gauge field is treated as background, not fluctuating, so there is a fibering of spaces of fields

$$\pi\colon \mathcal{F}_{\text{classical}} \longrightarrow \mathcal{F}_{\text{quantum}}, \tag{3.1}$$

with fiber the gauge field. The anomaly is classical in the sense that it is computed directly in the classical theory,[11] but it does not depend on the gauge field so is the pullback of an anomaly on $\mathcal{F}_{\text{quantum}}$. In this situation we use the term ' 't Hooft anomaly'; the anomaly does not obstruct the path integral over the gauge field.

## 3.1 The $SU(2)$ Theory

We begin by enumerating the relevant characteristic classes. First, identify the adjoint group[12] $PSU(2) \cong SO(3)$ and let

$$\rho\colon BSU(2) \longrightarrow BSO(3) \tag{3.2}$$

be the map on classifying spaces induced by the adjoint homomorphism $SU(2) \to SO(3)$. The main characters in the story are

$$\begin{aligned}
c_2 &\in H^4\big(BSU(2); \mathbb{Z}\big), \\
p_1 &\in H^4\big(BSO(3); \mathbb{Z}\big), \\
w_2 &\in H^2\big(BSO(3); \mathbb{Z}/2\mathbb{Z}\big),
\end{aligned} \tag{3.3}$$

which are, respectively, the second Chern class, the first Pontrjagin class, and the second Stiefel-Whitney class. They satisfy

$$\rho^*(p_1) = -4c_2 \tag{3.4}$$

$$p_1 \equiv \mathcal{P}(w_2) \pmod 4. \tag{3.5}$$

One can derive (3.4) using a technique pioneered in [45]. Namely, it suffices to compute for an $SU(2)$-bundle $L \oplus L^{-1}$ which is the sum of complex line bundles. Then the complexified adjoint bundle is $L^{\otimes 2} \oplus \underline{\mathbb{C}} \oplus L^{\otimes(-2)}$, where $\mathbb{C}$ denotes the trivial line bundle. Let $c(L) = 1 + x$ be the total Chern class. Then by the Whitney formula[13]

$$c(L \oplus L^{-1}) = c(L)c(L^{-1}) = (1+x)(1-x) = 1 - x^2 \tag{3.6}$$

and

$$c(L^{\otimes 2} \oplus \underline{\mathbb{C}} \oplus L^{\otimes(-2)}) = (1+2x)1(1-2x) = 1 - 4x^2. \tag{3.7}$$

We use the convention that $p_1$ of a real vector bundle is $-c_2$ of its complexification.

In (3.5) the operation

$$\mathcal{P}\colon H^2(-; \mathbb{Z}/2\mathbb{Z}) \longrightarrow H^4(-; \mathbb{Z}/4\mathbb{Z}) \tag{3.8}$$

is the Pontrjagin square. One derivation of (3.5) introduces the Lie group $U(2)$, whose adjoint group is also $SO(3)$. Arguing as above, with $L_1 \oplus L_2$ in place of $L \oplus L^{-1}$ and adjoint bundle $L_1 \otimes L_2^{-1} \oplus \underline{\mathbb{C}} \oplus L_1^{-1} \otimes L_2$, we deduce that under

$$\tilde{\rho}\colon BU(2) \longrightarrow BSO(3) \tag{3.9}$$

---

[11]as opposed, say, to the anomaly of a fermionic field, which arises from the fermionic path integral.

[12]The adjoint group of a connected Lie group $G$ is the quotient $G/Z$ by the center $Z$.

[13]Products in subsequent formulas are cup products unless otherwise indicated.

we have

$$\tilde{\rho}^*(p_1) = c_1^2 - 4c_2. \tag{3.10}$$

Hence $\mathrm{tr}^*(p_1) \equiv c_1^2 \pmod 4$. Now we need only use the relation

$$c_1 \equiv w_2 \pmod 2 \tag{3.11}$$

in $H^2\big(BU(2); \mathbb{Z}/2\mathbb{Z}\big)$ along with the fact that if $\tilde{x} \in H^2(-;\mathbb{Z})$ is an integral lift of $x \in H^2(-;\mathbb{Z}/2\mathbb{Z})$, then $\mathcal{P}(x) \equiv \tilde{x}^2 \pmod 4$.

With these preliminaries understood, we write the $\theta$-term (2.1) in the exponentiated action of 4d $SU(2)$ Yang-Mills theory as

$$\exp\big(-\sqrt{-1}\,\theta c_2(P)[M]\big), \tag{3.12}$$

where the gauge field is a connection $\Theta$ on a principal $SU(2)$-bundle $P \to M$ over a closed oriented 4-manifold $M$ with fundamental homology class $[M]$. (The transition from (3.12) to (2.1) is via Chern-Weil theory.) This expression is valid for a constant $\theta \in \mathbb{R}/2\pi\mathbb{Z}$.

Our first generalization of (3.12) is for $\theta\colon M \to \mathbb{R}/2\pi\mathbb{Z}$ a smooth function, not constrained to be locally constant. In this case we bring in differential cohomology, as reviewed in [4, §5] and the references therein. The function $\theta$ and connection $\Theta$ determine differential cohomology classes

$$\begin{aligned}
[\theta] &\in \check{H}^1(M; 2\pi\mathbb{Z}), \\
[\check{c}_2(\Theta)] &\in \check{H}^4(M; \mathbb{Z}),
\end{aligned} \tag{3.13}$$

where $\check{c}_2$ is the differential lift of the second Chern class, an amalgam of the Chern-Weil form and Chern-Simons form introduced by Cheeger-Simons [46]. The generalization of (3.12) is

$$\exp\left(-\sqrt{-1} \int_M [\theta] \cdot [\check{c}_2(\Theta)]\right), \tag{3.14}$$

where the dot is the product in differential cohomology.

*Remark* 3.15. If we drop the orientation on $M$, then $\theta\colon M_{w_1} \to \mathbb{R}/2\pi\mathbb{Z}$ is a function on the total space of the orientation double cover $M_{w_1} \to M$, and we require $\sigma^*\theta = -\theta$ for the non-identity deck transformation $\sigma\colon M_{w_1} \to M_{w_1}$. The class $[\theta] \in \check{H}^1(M; 2\pi\mathbb{Z}_{w_1})$ has coefficients in the local system defined by the orientation double cover. Passing to unoriented manifolds implements time-reversal symmetry. The only *constant* values of $\theta$ are then $\theta = 0$ and $\theta = \pi$.

*Remark* 3.16. The exponentiated $\theta$-term is the partition function of an invertible field theory. As such it has a (universal) expression analogous to that in [4, Example 6.14].

Pure $SU(2)$ Yang-Mills theory has a symmetry group[14] $B\mu_2$, elsewhere called the "center 1-form symmetry" attached to the center $\mu_2 \subset SU(2)$. The symmetry uses the homomorphism $\mu_2 \times SU(2) \to SU(2)$ to construct a new $SU(2)$-connection which "tensors" a double cover with $\Theta$. We seek to extend the theory to include a background field for the $B\mu_2$-symmetry, i.e., a $\mu_2$-gerbe. The latter can be thought of as a map $B\colon M \to B^2\mu_2$ with homotopy class $[B] \in H^2(M; \mu_2)$, the isomorphism class of the $\mu_2$-gerbe. As explained in §2.1 and previous references, the gauge field $\Theta$ is now a connection on a principal $SO(3)$-bundle over $M$. The situation is summarized in a universal fibering (see [47])

$$B_{\triangledown}SU(2) \longrightarrow B_{\triangledown}SO(3) \longrightarrow B^2\mu_2. \tag{3.17}$$

---

[14]The group $\mu_N = \{e^{2\pi ik/N} : k = 0, 1, \ldots, N-1\}$ of complex $N^{\mathrm{th}}$ roots of unity is the center of $SU(N)$. Elsewhere we denote the center of $SU(N)$ as '$\mathbb{Z}_N$'.

The expression (3.14) is well-defined for $\Theta$ in the fiber; our task is to extend it to the entire space.

From (3.4) it is clear that we would like to write

$$\text{``}\exp\left(\sqrt{-1}\int_M [\theta]\cdot[\check{p}_1(\Theta)/4]\right)\text{''}. \tag{3.18}$$

However, this does not make sense as a number: we cannot divide an integer by 4 inside the integers. The same prohibition applies to (differential) cohomology classes with integral coefficients. Note that this issue already occurs for constant $\theta$ (and oriented manifolds). It is this division by 4 which introduces an anomaly. The anomaly is easily computable from the well-defined expression

$$\exp\left(\sqrt{-1}\int_M [\theta]\cdot[\check{p}_1(\Theta)]\right), \tag{3.19}$$

whose 4$^{\text{th}}$ root we seek. We pause to explain root extraction in general terms.

First, let $S$ be a smooth manifold and $f : S \to \mathbb{C}^\times$ a smooth function to the nonzero complex numbers. A 4$^{\text{th}}$ root of $f$ is a lift to a function indicated by the dotted arrow in the diagram

$$\begin{array}{ccc} & & \mathbb{C}^\times \\ & \nearrow & \downarrow {\scriptstyle (-)^4} \\ S & \xrightarrow{\ f\ } & \mathbb{C}^\times \end{array} \tag{3.20}$$

However, there is an obstruction to its existence. Instead, form the pullback square

$$\begin{array}{ccc} \widetilde{S} & \xrightarrow{\ \tilde{f}\ } & \mathbb{C}^\times \\ {\scriptstyle \pi}\downarrow & & \downarrow {\scriptstyle (-)^4} \\ S & \xrightarrow{\ f\ } & \mathbb{C}^\times \end{array} \tag{3.21}$$

with

$$\widetilde{S} = \left\{(s,\lambda)\in S\times\mathbb{C}^\times : f(s)=\lambda^4\right\}. \tag{3.22}$$

Then $\pi$ is a 4-fold covering. The map $\tilde{f}$ is the canonical 4$^{\text{th}}$ root of $f$. However, it is a function on $\widetilde{S}$, not on $S$. There is a free $\mu_4$-action on $\widetilde{S}$ and a (nonfree) $\mu_4$-action on $\mathbb{C}^\times$; the map $\tilde{f}$ is equivariant. So $\tilde{f}$ descends to $S$, not as a $\mathbb{C}^\times$-valued function, but as a section of a line bundle $L_f \to S$. The line bundle has order 4 in the sense that it comes with a canonical trivialization of $L_f^{\otimes 4} \to S$.

Therefore, for a single manifold $M$ equipped with $\theta,\Theta$, the desired exponentiated $\theta$-term (3.18) is a well-defined element of a complex line of order 4. This indicates that the extended theory with the $B$-field is anomalous; the anomaly evaluated on $(M,\theta,\Theta)$ is the line.

We can also analyze division by 4 in differential cohomology. Ignoring factors of $2\pi$ for the moment, the short exact sequence of abelian groups

$$0 \longrightarrow \mathbb{Z} \xrightarrow{\ 4\ } \mathbb{Z} \longrightarrow \mathbb{Z}/4\mathbb{Z} \longrightarrow 0 \tag{3.23}$$

induces a long exact sequence

$$\cdots \longrightarrow H^4(-;\mathbb{Z}/4\mathbb{Z}) \longrightarrow \check{H}^5(-;\mathbb{Z}) \xrightarrow{\ 4\ } \check{H}^5(-;\mathbb{Z}) \longrightarrow H^5(-;\mathbb{Z}/4\mathbb{Z}) \longrightarrow \cdots . \tag{3.24}$$

Hence division by 4 of the product $[\theta]\cdot[\check{p}_1(\Theta)]$ is obstructed by a class in $H^5(-;\mathbb{Z}/4\mathbb{Z})$. Evaluate that class on the total space $\mathcal{M}\to S$ of a fiber bundle of triples $(M,\theta,\Theta)$ and integrate over

the fibers. The result is the isomorphism class of $\alpha(\mathcal{M} \to S, \theta, \Theta)$ in $H^1(S; \mathbb{Z}/4\mathbb{Z})$, where $\alpha$ is the anomaly theory. (The value $\alpha(\mathcal{M} \to S, \theta, \Theta)$ of the anomaly theory on a family of closed 4-manifolds is a flat complex line bundle of order 4.) If the principal $SO(3)$-bundle underlying the connection $\Theta$ is $Q \to \mathcal{M}$, then the result is

$$\int_{\mathcal{M}/S} [\theta] \cup [p_1(Q)] \quad (\text{mod } 4), \tag{3.25}$$

where now $[\theta] \in H^1(\mathcal{M}; 2\pi\mathbb{Z})$ is the topological cohomology class of $\theta$. By (3.5) this equals

$$\int_{\mathcal{M}/S} [\theta] \cup \mathcal{P}(B) \tag{3.26}$$

(Compare (2.5)).

*Remark* 3.27. The same computation can be done universally, as in [4, §7], to deduce the isomorphism class of the anomaly theory $\alpha$, which is a 5d topological field theory of order 4. The result, as in (3.26), depends only on the background fields in $\mathcal{F}_{\text{quantum}}$; see (3.1).

*Remark* 3.28. The same formula (3.26) works when we include time-reversal symmetry by dropping the orientation and defining $\theta$ as twisted by the orientation double cover. We can then specialize to $\theta = \pi$, as in the last paragraph of [4, §7.1].

## 3.2 More General Lie Groups

Now consider 4$d$ Yang-Mills theory with gauge group $G$. Assume $G$ is compact and connected, and let $\Gamma \subset G$ be a finite subgroup of the center of $G$. (We can contemplate more general situations, but here confine ourselves to these hypotheses.) A $\theta$-term (3.12) or (3.14) can be defined[15] for every integral characteristic class $\lambda \in H^4(BG; \mathbb{Z})$, in some contexts called a *level* of $G$. The group $B\Gamma$ acts as a ("1-form") symmetry. Therefore, we ask to extend the theory to include a background $\Gamma$-gerbe $B$, whose isomorphism class $[B]$ lies in $H^2(-; \Gamma)$.

*Remark* 3.29. More generally, if $\lambda_1, \dots, \lambda_k \in H^4(BG; \mathbb{Z})$ are linearly independent over $\mathbb{Q}$, then we introduce $\mathbb{R}/2\pi\mathbb{Z}$-valued functions $\theta^1, \dots, \theta^k$ and put the sum $\theta^i \lambda_i$ in place of $\theta c_2$ in (3.12). Example: $G = U(N)$, $\lambda_1 = c_1^2$, $\lambda_2 = c_2$. To keep the presentation simpler, we proceed with $k = 1$.

Set $\overline{G} = G/\Gamma$. The relevant fibering of fields is

$$B_\nabla G \longrightarrow B_\nabla \overline{G} \longrightarrow B^2\Gamma. \tag{3.30}$$

The second map sends a $\overline{G}$-connection to a $\Gamma$-gerbe, the obstruction to lifting it to a $G$-connection. The original theory is defined for $G$-connections and we seek to extend to $\overline{G}$-connections.

The extension problem is topological, since $\Gamma$ is assumed finite, so we can drop the connections and replace (3.30) by the fibration

$$BG \longrightarrow B\overline{G} \longrightarrow B^2\Gamma \tag{3.31}$$

of topological spaces. Our problem is to extend $\lambda$ to a cohomology class on $B\overline{G}$. In Algebraic Topology one analyzes this extension problem cell-by-cell for a CW structure on $B^2\Gamma$; the computation is encoded in the Leray-Serre spectral sequence of (3.30).

---

[15]The main theorem in [48] implies that $\lambda$ has a unique lift to a universal differential characteristic class $\check{\lambda} \in \check{H}^4(B_\nabla G; \mathbb{Z})$ for $G$-connections.

**Example 3.32.** In §3.1 we have $G = SU(2)$, $\Gamma = \mu_2$, $\overline{G} = SO(3)$, and $\lambda = -c_2$. The class $\lambda$ does not extend to $BSO(3)$. Rather, it extends over the inverse image of the 4-skeleton of $B^2\mu_2$, but then one hits an obstruction on the 5-skeleton. In other words, $\lambda \in H^4(BG;\mathbb{Z})$ *transgresses* to a class[16] in $H^5(B^2\Gamma;\mathbb{Z})$. In this situation—obstruction only at the last stage—there is an anomalous extension with an anomaly theory in ordinary Eilenberg-MacLane cohomology.

*Remark* 3.33. If one hits obstructions at earlier stages, there may still be an anomalous extension, but with anomaly theory described by a more complicated spectrum.

**Example 3.34.** Let $G = SU(N)$, set $\Gamma = \mu_N$ the center of $G$, and then $\overline{G} = PSU(N)$. Fix $\lambda = -c_2 \in H^4(BG;\mathbb{Z})$. Now both $H^4(BG;\mathbb{Z})$ and $H^4(B\overline{G};\mathbb{Z})$ have rank 1, and the pullback map $\rho^* \colon H^4(B\overline{G};\mathbb{Z}) \to H^4(BG;\mathbb{Z})$ is injective, but not an isomorphism. Choose the generator $\bar\lambda \in H^4(B\overline{G};\mathbb{Z})$ such that[17] $\rho^*(\bar\lambda) = m(-c_2)$ for $m \in \mathbb{Z}^{>0}$. The value of $m$ was calculated in (2) below Lemma 4.7 in [49] as

$$m = m(N) = \begin{cases} 2N, & N \text{ even;} \\ N, & N \text{ odd} \end{cases} \tag{3.35}$$

(We prove (3.35) by an alternative method below). To compute the anomaly, we need the mod $m$ reduction of $\bar\lambda$. Let $w \in H^2(B\overline{G};\mathbb{Z}/N\mathbb{Z})$ be the lift of the tautological class $\iota \in H^2(B^2\mu_N;\mathbb{Z}/N\mathbb{Z})$ under an identification[18] $\mu_N \cong \mathbb{Z}/N\mathbb{Z}$; then $w$ is the obstruction to lifting a $\overline{G}$-bundle to a $G$-bundle, which deserves the moniker 'Brauer class'. Set $\widetilde{G} = U(N)$ and observe that $\overline{G} = PSU(N) = PU(N)$ is also its adjoint group. The abelian group $H^4(B\widetilde{G};\mathbb{Z})$ is free of rank 2 with generators $c_1^2, c_2$. Let $\tilde\rho \colon B\widetilde{G} \to B\overline{G}$ be the map induced from the adjoint homomorphism $\widetilde{G} \to \overline{G}$. Then

$$\tilde\rho^*(w) = c_1 \pmod{N} \tag{3.36}$$

fixes the generator $w$ (see footnote [18]). Gu in [49] computes a more precise formula than (3.35), namely

$$\tilde\rho^*(\bar\lambda) = \begin{cases} (N-1)c_1^2 - 2Nc_2, & N \text{ even;} \\ \frac{1}{2}\big((N-1)c_1^2 - 2Nc_2\big), & N \text{ odd,} \end{cases} \tag{3.37}$$

a formula we reproduce below. It follows from (3.36) and (3.37) that

$$\bar\lambda \equiv \begin{cases} (N-1)\mathcal{P}(w) \pmod{2N}, & N \text{ even;} \\ \frac{N-1}{2}\, w \cup w \pmod{N}, & N \text{ odd.} \end{cases} \tag{3.38}$$

Arguing as in (3.26), we conclude that the partition function of the anomaly theory $\alpha$ on a closed 5-manifold $W$ with $\mu_N$-gerbe $B$ and function $\theta \colon W \to \mathbb{R}/2\pi\mathbb{Z}$ is the exponential of $\sqrt{-1}$ times

$$\begin{cases} (N-1)[\theta] \cup \mathcal{P}(w), & N \text{ even;} \\ \frac{N-1}{2}[\theta] \cup w \cup w & N \text{ odd} \end{cases} \tag{3.39}$$

(Compare (2.5)).

Next, we introduce a device we apply to deduce (3.37). Let $\widetilde{G}$ be a compact connected Lie group and $\overline{G}$ the quotient by a subgroup of the center. Assume $H^4(B\overline{G};\mathbb{Z})$ has rank 1, so the pullback

$$\tilde\rho^* \colon H^4(B\overline{G};\mathbb{Z}) \longrightarrow H^4(B\widetilde{G};\mathbb{Z}) \tag{3.40}$$

---

[16]It is $\beta_{\mathbb{Z}}\mathcal{P}(\iota)$, where $\iota \in H^2(B^2\Gamma;\Gamma)$ is the tautological class and $\beta_{\mathbb{Z}}$ the Bockstein induced by the short exact sequence $0 \to \mathbb{Z} \xrightarrow{4} \mathbb{Z} \to \mathbb{Z}/4\mathbb{Z} \to 0$.

[17]For $N = 2$ we have $\bar\lambda = p_1 \in H^4(BSO(3);\mathbb{Z})$; see (3.4).

[18]We fix the indeterminacy below in (3.36).

has image a rank 1 sublattice. We want to determine $\tilde{\rho}^*(\bar{\lambda})$ for $\bar{\lambda}$ a generator of $H^4(B\overline{G}; \mathbb{Z})$. Let

$$\mathrm{Ad}_{\mathbb{C}} \colon \widetilde{G} \longrightarrow \overline{G} \longrightarrow SO(\mathfrak{g}) \longrightarrow SU(\mathfrak{g}) \tag{3.41}$$

be the complexified adjoint representation of $\widetilde{G}$ on its Lie algebra $\mathfrak{g}$. Then $c_2(\mathrm{Ad}_{\mathbb{C}}) \in H^4(B\widetilde{G}; \mathbb{Z})$ lies in the image of $\tilde{\rho}^*$, so is an integer multiple of $\tilde{\rho}^*(\bar{\lambda})$. In fortuitous circumstances we may be able to prove that the integer is $\pm 1$. We compute $c_2(\mathrm{Ad}_{\mathbb{C}})$ by the technique of [45]. Let $\widetilde{T} \subset \widetilde{G}$ be a maximal torus. The restriction of $\mathrm{Ad}_{\mathbb{C}}$ to $\widetilde{T}$ reduces to a sum of 1-dimensional representations with characters labeled by the set $\Delta$ of (infinitesimal) roots of $\widetilde{G}$ plus a trivial representation of dimension rank $\widetilde{G}$. Order the elements of $\Delta$. Then

$$c_2(\mathrm{Ad}_{\mathbb{C}}) = \sum_{\substack{\alpha < \beta \\ \alpha, \beta \in \Delta}} \alpha\beta. \tag{3.42}$$

Here we identify roots as elements of the character lattice $H^2(B\widetilde{T}; \mathbb{Z}) \cong \mathrm{Hom}(\widetilde{T}, \mathbb{T})$, and (3.42) lies in the image of $H^4(B\widetilde{G}; \mathbb{Z}) \to H^4(B\widetilde{T}; \mathbb{Z})$. Choose a Weyl chamber and so a partition $\Delta = \Delta^+ \amalg -\Delta^+$. An easy manipulation which begins by squaring $\sum_{\alpha \in \Delta} \alpha = 0$ proves

$$c_2(\mathrm{Ad}_{\mathbb{C}}) = -\sum_{\alpha \in \Delta^+} \alpha^2. \tag{3.43}$$

**Example 3.44.** We apply (3.43) to prove (3.37); then (3.35) follows by restriction along $BSU(N) \to BU(N)$. As in Example 3.34 set $G = SU(N)$, $\widetilde{G} = U(N)$, $\overline{G} = PSU(N)$. Let $\widetilde{T} \subset \widetilde{G}$ be the standard maximal torus of diagonal matrices and $x_i$ the (infinitesimal) character obtained by projection onto the $i^{\mathrm{th}}$ diagonal entry. With the appropriate Weyl chamber $\Delta^+ = \{x_i - x_j : 1 \leqslant i < j \leqslant N\}$. So

$$\begin{aligned}
c_2(\mathrm{Ad}_{\mathbb{C}}) &= -\sum_{1 \leqslant i < j \leqslant N} (x_i - x_j)^2 \\
&= -(N-1)\sum_{i=1}^{N} x_i^2 + 2\sum_{1 \leqslant i < j \leqslant N} x_i x_j \\
&= -(N-1)\left(\sum_{i=1}^{N} x_i\right)^2 + 2N\sum_{1 \leqslant i < j \leqslant N} x_i x_j \\
&= -(N-1)c_1^2 + 2Nc_2.
\end{aligned} \tag{3.45}$$

If $N$ is even, then this class is primitive so must be $\pm\tilde{\rho}(\bar{\lambda})$ for a generator $\bar{\lambda} \in H^4(B\overline{G}; \mathbb{Z})$. If $N$ is odd, then $c_2(\mathrm{Ad}_{\mathbb{C}})$ is divisible by 2. Without further information we do not know if the corresponding class $\bar{\mu} \in H^4(B\overline{G}; \mathbb{Z})$ is divisible by 2. (Here $\bar{\mu}$ is $c_2 2$ of the complexified adjoint representation of $\overline{G}$.) We argue in the affirmative as follows. In the fibration (3.31) with $G = SU(N)$, the Eilenberg-MacLane space $B^2\mu_N$ has trivial mod 2 cohomology, since $N$ is odd, and so pullback $\rho^* \colon H^4(B\overline{G}; \mathbb{Z}/2\mathbb{Z}) \to H^4(BG; \mathbb{Z}/2\mathbb{Z})$ is an isomorphism. Since $\rho^*(\bar{\mu}) = 2Nc_2$ is divisible by 2, so is $\bar{\mu}$. This completes the proof of (3.37).

# 4   4$d$ QCD

In this section we consider 4$d$ QCD with fermions. Specifically, we will study $SU(N)$ and $Sp(N)$ with matter in the fundamental representation. This means that these theories do not have any one-form global symmetry.

Despite the absence of a one-form symmetry, these systems can still have a mixed anomaly between the $\theta$-periodicity and its global symmetry. The reason is that even without a one-form global symmetry, twisted bundles of the dynamical gauge fields can be present with appropriate background of the gauge fields of the zero-form global symmetry.[19] These bundles do not have integer instanton numbers and hence they lead to our anomaly.

As we will see, even when all possible bundles of the dynamical fields can be present, the anomaly is not the same as in the corresponding gauge theory without matter in section 2. Some of that putative anomaly can be removed by adding appropriate counterterms.

This discussion extends the results about interfaces in $4d$ in [26] and explains the relation between them and the earlier results about anomalies in $3d$ Chern-Simons-matter theory in [15].

To briefly summarize our results, we will find that the $SU(N)$ theory with $N_f$ fundamental quarks has a non-trivial anomaly (4.17) if and only if $L = \gcd(N, N_f) > 1$. Meanwhile the $Sp(N)$ theory with $N_f$ fundamental quarks has a non-trivial anomaly (4.38) if and only if $N$ is odd and $N_f$ is even. We interpret these results in terms of the dynamics of the Chern-Simons matter theories that reside on their interfaces.

## 4.1 $SU(N)$ QCD

We begin with $4d$ $SU(N)$ QCD with $N_f$ fermions in the fundamental representation. The Euclidean action is

$$S = \int -\frac{1}{4g^2}\text{Tr}(f \wedge *f) - \frac{i\theta}{8\pi^2}\text{Tr}(f \wedge f) + i\overline{\psi}_I \slashed{D}_a \psi^I + i\overline{\widetilde{\psi}}_I \slashed{D}_a \widetilde{\psi}^I + (m\widetilde{\psi}_I \psi^I + c.c.), \quad (4.1)$$

where $f$ is the field strength of the $SU(N)$ gauge field $a$. Here we suppressed the color indices and used the standard summation convention for the flavor indices $I$. The theory only depends on the complex parameter $me^{i\theta/N_f}$, so without lost of generality we will take $m$ to be a positive real parameter. Since the theory contains fermions, we will limit ourselves to spin manifolds, even though with an appropriate twist the theory can be placed on certain non-spin manifolds.

With equal masses the global symmetry of the system that acts faithfully is

$$\mathcal{G} = \frac{U(N_f)}{\mathbb{Z}_N} . \quad (4.2)$$

To see that, note that locally the fermions transform under

$$\mathcal{G}'_{micro} = SU(N) \times SU(N_f) \times U(1) , \quad (4.3)$$

where the first factor is the gauge group, the second factor is the flavor group, and the $U(1)$ is the baryon number normalized to have charge one for the fundamental quarks. However, $\mathcal{G}'_{micro}$ does not act faithfully on the quarks. The group that acts faithfully on them is

$$\mathcal{G}_{micro} = \frac{\mathcal{G}'_{micro}}{\mathbb{Z}_N \times \mathbb{Z}_{N_f}} = \frac{SU(N) \times U(N_f)}{\mathbb{Z}_N} . \quad (4.4)$$

Here $\mathcal{G}'_{micro}$ is represented by $(u \in SU(N), v \in SU(N_f), w \in U(1))$ and the quotient is the identification

$$(u, v, w) \sim (e^{2\pi i/N}u, v, e^{-2\pi i/N}w) \sim (u, e^{2\pi i/N_f}v, e^{-2\pi i/N_f}w) . \quad (4.5)$$

Finally, the global symmetry $\mathcal{G} = U(N_f)/\mathbb{Z}_N$ (4.2) is obtained by moding out $\mathcal{G}_{micro}$ by the $SU(N)$ gauge group.

---

[19]Many people have studied twisted bundles of the dynamical fields using a twist in the flavor to compensate it. For an early paper, see e.g. [50]. For more recent related discussions in $4d$ see [26, 51–57] and in $3d$ see [14, 15] and references therein.

### 4.1.1 Anomalies Involving $\theta$-periodicity

In order to study the anomaly, we should couple the global symmetry $\mathcal{G} = U(N_f)/\mathbb{Z}_N$ (4.2) to background gauge fields. We will do it in steps. First, we couple the theory to $SU(N_f) \times U(1)$ background gauge fields $(A, C)$ (the fundamental fermions have charge one under the $U(1)$). Together with the dynamical $SU(N)$ gauge fields $a$ these gauge fields represent $\mathcal{G}'_{micro}$ (4.3).

Next, we would like to perform the quotient leading to $\mathcal{G}_{micro}$ (4.4). We do that by letting $a$ be a $PSU(N)$ gauge field, $A$ be a $PSU(N_f)$ gauge field, and $\widetilde{C} = KC$ with $K = \text{lcm}(N, N_f)$ be a $\widetilde{U}(1) = U(1)/\mathbb{Z}_K$ gauge field. Then, the gauge fields $(a, A, \widetilde{C})$ are correlated through

$$\oint \frac{\widetilde{F}}{2\pi} = \oint \left( \frac{N_f}{L} w_2(a) + \frac{N}{L} w_2(A) \right) \text{ mod } K , \tag{4.6}$$

where $\widetilde{F} = d\widetilde{C}$,

$$K = \text{lcm}(N, N_f), \qquad L = \gcd(N, N_f) = \frac{N N_f}{K} \tag{4.7}$$

and $w_2$ is the second Stiefel-Whitney class of the corresponding bundles.

In terms of these gauge fields, the background fields for $\mathcal{G} = U(N_f)/\mathbb{Z}_N$ are $A$ and $\widetilde{C}$ in $PSU(N_f) \times \widetilde{U}(1)$ constrained to satisfy

$$\frac{L}{N_f} \oint \frac{\widetilde{F}}{2\pi} = \frac{N}{N_f} \oint w_2(A) \text{ mod } 1 . \tag{4.8}$$

Arbitrary values of these gauge fields, subject to (4.8), allow us to probe arbitrary values of $w_2(a)$ for the dynamical gauge fields. It is determined by a class $w_2^{(N)} \in H^2(X, \mathbb{Z}_N)$ of the $\mathcal{G} = U(N_f)/\mathbb{Z}_N$ gauge fields $A$ and $\widetilde{C}$, which represents the obstruction to it being a $U(N_f)$ gauge field. Specifically,

$$\oint w_2(a) = \oint w_2^{(N)} = \left( \frac{L}{N_f} \oint \frac{\widetilde{F}}{2\pi} - \frac{N}{N_f} \oint w_2(A) \right) \text{ mod } N . \tag{4.9}$$

Note that $w_2^{(N)}$ depends only on the background fields.

Now that we can use the background fields to induce arbitrary $w_2(a)$ we can repeat the analysis in section 2 to find that under shifting $\theta \to \theta + 2\pi$ the action is shifted

$$-\frac{2\pi i}{N} \int \frac{\mathcal{P}(w_2(a))}{2} = -\frac{2\pi i}{N} \int \frac{\mathcal{P}(w_2^{(N)})}{2} \text{ mod } 2\pi i , \tag{4.10}$$

where in the last expression we expressed it in terms of the background fields $A$ and $\widetilde{C}$ as in (4.9), showing that it is an anomaly.

We might be tempted to interpret (4.10) as a $\mathbb{Z}_N$ anomaly. However, this is not the case.

To see that, we proceed as follows. Using (4.6) and (4.9) it is straightforward to check that

$$\begin{aligned}
\exp\left( -\frac{2\pi i}{N} \int \frac{\mathcal{P}(w_2^{(N)})}{2} \right) &= \exp\left( \frac{2\pi i}{L} \int \left( R \frac{\mathcal{P}(w_2^{(N)})}{2} + J w_2^{(N)} \cup w_2(A) \right) \right) \\
&\quad \exp\left( 2\pi i \int \left( -\frac{J}{K} \frac{\widetilde{F} \wedge \widetilde{F}}{8\pi^2} + \frac{NJ}{L} \frac{\mathcal{P}(w_2(A))}{2N_f} \right) \right),
\end{aligned} \tag{4.11}$$

where $R$ and $J$ are integers satisfying

$$J N_f - R N = L \tag{4.12}$$

(Different solutions of this equation for $(J, R)$ lead to the same value in (4.11)). The significance of the apparently unmotivated expression (4.11) will be clear soon.

Given that we have background fields $A$ and $C$, we can add some counterterms to the action. Two special terms are

$$\frac{i\Theta_A}{8\pi^2} \int \mathrm{Tr}(F_A \wedge F_A) + \frac{i\Theta_C}{8\pi^2} \int F_C \wedge F_C \,. \tag{4.13}$$

The normalization here is such that for an $SU(N_f) \times U(1)$ background $(A, C)$ the coefficients $\Theta_A$ and $\Theta_C$ are $2\pi$-periodic.

The new crucial point is that when we study the anomaly in the shift of $\theta$ we can combine this operation with continuous shifts of $\Theta_A$ and $\Theta_C$. In other words, we can think of $\Theta_A$ and $\Theta_C$ as being $\theta$-dependent[20]

$$\Theta_A = \Theta_A^{(0)} + n_A \theta \,, \qquad \Theta_C = \Theta_C^{(0)} + n_C \theta \,. \tag{4.14}$$

In order to preserve the $2\pi$-periodicity of $\theta$ for $SU(N_f) \times U(1)$ background fields we take $n_A, n_C \in \mathbb{Z}$. Then, under $\theta \to \theta + 2\pi$ the expression (4.13) is shifted by (recall that $\widetilde{C} = KC$)

$$2\pi i \int \left( n_A \frac{\mathrm{Tr}(F_A \wedge F_A)}{8\pi^2} + n_C \frac{\widetilde{F} \wedge \widetilde{F}}{8\pi^2 K^2} \right) = 2\pi i \int \left( -n_A \frac{\mathcal{P}(w_2(A))}{2N_f} + n_C \frac{\widetilde{F} \wedge \widetilde{F}}{8\pi^2 K^2} \right) \bmod 2\pi i \,. \tag{4.15}$$

Comparing this with (4.11) we see that by choosing

$$n_A = \frac{N}{L} J \,, \qquad n_C = JK \tag{4.16}$$

(note that $N/L$ is an integer), we can cancel the second factor in (4.11). This leaves us with an anomaly only because of the first factor in (4.11). As in all the examples above, it can be written as a $5d$ anomaly action

$$\mathcal{A}(\theta, A, C) = \exp\left( \frac{2\pi i}{L} \int \frac{d\theta}{2\pi} \left( R \frac{\mathcal{P}(w_2^{(N)})}{2} + J w_2^{(N)} \cup w_2(A) \right) \right) \,. \tag{4.17}$$

It is crucial that unlike the variation (4.10), which appears to be a $\mathbb{Z}_N$ anomaly, this expression is only a $\mathbb{Z}_L$ anomaly.

Finally, we show that using the freedom in (4.14), we cannot remove this $\mathbb{Z}_L$ anomaly. In other words, we show that there are no integer shifts of $n_A$ and $n_C$ in (4.15) that can make the partition function invariant under $\theta \to \theta + 2\pi r$ with $r \neq 0 \bmod L$. We try to satisfy

$$
\begin{aligned}
\frac{r}{N} \int \frac{\mathcal{P}(w_2^{(N)})}{2} &= \left( \frac{s}{8\pi^2} \int \mathrm{Tr}(F_A \wedge F_A) + \frac{t}{8\pi^2 K^2} \int \widetilde{F} \wedge \widetilde{F} \right) \bmod 1 \\
&= \left( -\frac{s}{N_f} \int \frac{\mathcal{P}(w_2(A))}{2} + \frac{t}{8\pi^2 K^2} \int \widetilde{F} \wedge \widetilde{F} \right) \bmod 1 \,,
\end{aligned}
\tag{4.18}
$$

with integer $s$ and $t$. Clearly, we must have $t \in K\mathbb{Z}$. Then, using (4.6) it becomes

$$\left( \frac{(t - sN_f)}{N_f^2} \int \frac{\mathcal{P}(w_2(A))}{2} + \frac{t}{N^2} \int \frac{\mathcal{P}(w_2^{(N)})}{2} + \frac{t}{NN_f} \int w_2(A) \cup w_2^{(N)} \right) \bmod 1 \,. \tag{4.19}$$

---

[20]We could have added to (4.13) another linearly independent counterterm $\frac{2\pi i p}{N} \int \frac{\mathcal{P}(w_2^{(N)})}{2}$. However, since its coefficient $p$ is quantized, it cannot depend on $\theta$ as here and therefore it cannot be used to remove the variation in (4.11). This counterterm will be important in section 4.1.2.

Comparing with (4.18), we find that the coefficients $(s, t, r)$ should satisfy

$$t - sN_f \in N_f^2 \mathbb{Z}, \quad t - rN \in N^2 \mathbb{Z}, \quad t \in NN_f \mathbb{Z}. \tag{4.20}$$

These conditions can be satisfied only if $r = 0 \bmod L$. These manipulations are identical to the discussion in section 2.2 in [15]. The reason for this relation will be clear soon.

We conclude that our theory has the anomaly (4.17). As a result, the theory is invariant only under $\theta \to \theta + 2\pi L$ and the anomaly is absent when $L = \gcd(N, N_f) = 1$.

When $L = \gcd(N, N_f) \neq 1$, the anomaly prohibits the long distance theory to be trivially gapped everywhere between $\theta$ and $\theta + 2\pi$. For small enough $N_f$ the theory is believed to be trivially gapped at generic $\theta$ and nonzero mass. Therefore, the anomaly implies at least one phase transition when $\theta$ varies by $2\pi$. This is consistent with [26], where it was argued for different behavior depending on $N_f$ and the value of the mass.

The anomaly also constrains smooth interfaces between regions with different $\theta$. Suppose the two regions have $\theta$ and $\theta + 2\pi k$ for some integer $k$. The theory on the interface then has an ordinary 't Hooft anomaly of the zero-form global symmetry $U(N_f)/\mathbb{Z}_N$

$$\exp\left( \frac{2\pi i k}{L} \int \left( R \frac{\mathcal{P}(w_2^{(N)})}{2} + Jw_2(A) \cup w_2^{(N)} \right) \right). \tag{4.21}$$

It is trivial when $k = 0 \bmod L$.

Clearly, this anomaly does not uniquely determine the theory on the interface (see e.g. the related discussion in [37] and the comments below). One possible choice for the theory on the interface is the 3d Chern-Simons-matter theory[21]

$$SU(N)_{k-N_f/2} + N_f \text{ fermions} \tag{4.22}$$

or its dual theories

$$
\begin{array}{ll}
U(k)_{-N} + N_f \text{ scalars} & k \geqslant 1, \\
U(N_f - k)_N + N_f \text{ scalars} & k < N_f,
\end{array}
\tag{4.23}
$$

with a $U(N_f)$ invariant scalar potential. The fact that there are two dual scalar theories for $1 \leqslant k < N_f$ was important in [16]. See the discussion there for more details about the validity of these dualities. All these theories have a $U(N_f)/\mathbb{Z}_N$ global symmetry with the anomaly (4.21) [15]. In deriving this anomaly the freedom to add Chern-Simons counterterms of the background gauge fields was used. These Chern-Simons counterterms can be thought of as being induced by the continuous counterterms (4.13) in the 4d theory. This explains the relation between the computation of the anomaly under shifts of $\theta$ above with the computation of the anomaly in the 3d theory in section 2.2 in [15].

Further information about the theory along the interface can be found by considering the limits of large and small fermion masses. For $1 \leqslant N_f < N_{CFT}$ (with $N_{CFT}$ the lower boundary of the conformal window) the analysis of [26] showed that for $1 \leqslant k < N_f$ the theories (4.22)(4.23) indeed capture the phases of the interface theory. We will not repeat this discussion here.

### 4.1.2 Implications of Time-Reversal Symmetry

As we did in the previous examples, we would like to compare our discussion using the anomaly in shift of $\theta$ to what can be derived using ordinary anomalies of global symmetries involving time-reversal symmetry T (or equivalently a CP symmetry) at $\theta = 0, \pi$.

---

[21]The special case of $k = 1$ was discussed in detail in [26]. The generalization to larger $k$ was explored in appendix A of that paper.

Table 4: Summary of anomalies and existence of continuous counterterms preserving time-reversal symmetry T in 4$d$ QCD. Here $L = \gcd(N, N_f)$.

| theory | without T | with T at $\theta = 0, \pi$ | | |
|---|---|---|---|---|
| symmetry $\mathcal{G}$ | $\theta$-$\mathcal{G}$ anomaly | T-$\mathcal{G}$ anomaly at $\theta = \pi$ | no continuous counterterms | |
| $SU(N)$ QCD $\mathcal{G} = U(N_f)/\mathbb{Z}_N$ | even $L$ ✔ <br> odd $L \neq 1$ ✔ <br> $L = 1$ ✘ | even $L$ ✔ <br> odd $L \neq 1$ ✘ <br> $L = 1$ ✘ | even $L$ ✔ <br> odd $L \neq 1$ ✔ <br> $L = 1$ ✘ | |

First we discuss the possible counterterms that we can add to the theory. They are parameterized by[22]

$$\frac{\pi i s}{8\pi^2} \int \mathrm{Tr}(F_A \wedge F_A) + \frac{\pi i t}{8\pi^2 K^2} \int \widetilde{F} \wedge \widetilde{F} + \frac{2\pi i p}{N} \int \frac{\mathcal{P}(w_2^{(N)})}{2}. \tag{4.24}$$

All the other counterterms can be expressed as linear combinations of these three counterterms using (4.9). As in section 4.1.1, these counterterms have a redundancy which can be removed if we limit ourselves to $p$ mod $L$.

Now we discuss the T-symmetry at $\theta = \pi$. In order to preserve the T-symmetry in $SU(N_f) \times U(1)$ backgrounds (as opposed to more general $U(N_f)/\mathbb{Z}_N$ backgrounds), $s$ and $t$ have to be integers. Under the T-symmetry, the partition function transforms by

$$Z[\theta, A, \tilde{C}] \to Z[\theta, A, \tilde{C}] \exp\left( 2\pi i \int \left( (1 - 2p) \frac{\mathcal{P}(w_2^{(N)})}{2N} - s \frac{\mathrm{Tr}(F_A \wedge F_A)}{8\pi^2} - t \frac{\widetilde{F} \wedge \widetilde{F}}{8\pi^2 K^2} \right) \right). \tag{4.25}$$

Using the results in section 4.1.1, the transformations can be made non-anomalous with an appropriate choice of $s$ and $t$ if

$$1 - 2p = 0 \bmod L. \tag{4.26}$$

This equation has integer solutions for $p$ if $L$ is odd. Therefore, we conclude that the theory at $\theta = \pi$ has a mixed anomaly involving the time-reversal symmetry and the $U(N_f)/\mathbb{Z}_N$ zero-form symmetry only when $L = \gcd(N, N_f)$ is even. In that case, the theory at $\theta = \pi$ cannot be trivially gapped.

If $L = \gcd(N, N_f)$ is odd, the counterterms that preserve the T-symmetry at $\theta = 0$ and $\theta = \pi$ are different. In particular, we need to have $p = 0$ mod $L$ at $\theta = 0$ and $p = (L+1)/2$ mod $L$ at $\theta = \pi$. As with our various examples above, even though there is no anomaly for odd $L$, the fact that we need different counterterms at $\theta = 0$ and at $\theta = \pi$ can allow us to conclude that in that case the theory cannot be trivially gapped between $\theta = 0$ and $\theta = \pi$. There is an exception when $L = 1$. There we can choose $p = 0$ mod $L$ and find a continuous conterterm that preserves the T-symmetry at $\theta = 0, \pi$

$$i\theta \int \left( \frac{J}{NN_f} \frac{\widetilde{F} \wedge \widetilde{F}}{8\pi^2} + NJ \frac{\mathrm{Tr}(F_A \wedge F_A)}{8\pi^2} \right), \tag{4.27}$$

with an integer $J$ satisfying $JN_f = 1$ mod $N$.

The existence of continuous counterterms preserving the time-reversal symmetry T at $\theta = 0, \pi$ are summarized in Table 4.

---

[22]The discrete counterterm $\frac{2\pi i p}{N} \int \frac{\mathcal{P}(w_2^{(N)})}{2}$ was not included in [26]. Its significance will be clear below.

## 4.2 $Sp(N)$ **QCD**

Consider $Sp(N)$ QCD with $2N_f$ Weyl fermions in the fundamental $2N$-dimensional representation. Note that the theory is inconsistent with odd number of fermion multiplets due to a nonperturbative anomaly involving $\pi_4(Sp(N)) = \mathbb{Z}_2$ [58]. The Euclidean action includes the kinetic terms and

$$\mathcal{S} \supset -\frac{i\theta}{8\pi^2} \int \mathrm{Tr}(f \wedge f) + \int \left( m\Omega_{IJ}\widetilde{\Omega}^{ij}\psi_i^I\psi_j^J + c.c. \right), \tag{4.28}$$

where $\Omega_{IJ}$ and $\widetilde{\Omega}^{ij}$ are the invariant tensors of $Sp(N_f)$ and $Sp(N)$ respectively and we used the standard summation convention for the flavor indices $I, J$ and the color indices $i, j$. Note that we took equal masses $m$ for all the quarks. Because of the chiral anomaly, the theory depends only on the complex parameter $me^{i\theta/N_f}$, so without lost of generality we will take $m$ to be a positive real parameter. For simplicity, we will limit ourselves to spin manifolds.

With the $Sp(N_f)$ invariant mass term the faithful global symmetry of the system is

$$\mathcal{G} = \frac{Sp(N_f)}{\mathbb{Z}_2}. \tag{4.29}$$

To see that, note that locally the fermions transform under

$$\mathcal{G}'_{micro} = Sp(N) \times Sp(N_f), \tag{4.30}$$

where the first factor is the gauge group and the second factor is the flavor group. However the group that acts faithfully on the quarks is

$$G_{micro} = \frac{Sp(N) \times Sp(N_f)}{\mathbb{Z}_2}. \tag{4.31}$$

Here $G'_{micro}$ is represented by $(u \in Sp(N), v \in Sp(N_f))$ and the quotient is the identification

$$(u, v) \sim (-u, -v). \tag{4.32}$$

Finally the global symmetry $\mathcal{G} = Sp(N_f)/\mathbb{Z}_2$ (4.29) is obtained by moding out $\mathcal{G}_{micro}$ by the $Sp(N)$ gauge group.

### 4.2.1 Anomalies Involving $\theta$-periodicity

In order to study the anomaly, we couple the global symmetry $\mathcal{G} = Sp(N_f)/\mathbb{Z}_2$ (4.29) to background gauge field. We do it in steps. First, we couple the theory to $Sp(N_f)$ gauge field $A$. Together with the dynamical gauge field $a$, these gauge fields represent $\mathcal{G}'_{micro}$ (4.30).

Next we perform the quotient leading to $\mathcal{G}_{micro}$ (4.31), This promotes $a$ to be an $Sp(N)/\mathbb{Z}_2$ gauge field and $A$ to be an $Sp(N_f)/\mathbb{Z}_2$ gauge field. They are correlated via

$$w_2(a) = w_2(A), \tag{4.33}$$

where $w_2$ is the second Stiefel-Whitney class of the corresponding bundle.

As in the case of $SU(N)$ gauge theories above, we can use the background fields to induce arbitrary $w_2(a)$. Then, using (2.18), we find that shifting $\theta \to \theta + 2\pi$, the action is shifted by

$$2\pi i\frac{N}{2} \int \frac{\mathcal{P}(w_2(a))}{2} = 2\pi i\frac{N}{2} \int \frac{\mathcal{P}(w_2(A))}{2} \mod 2\pi i. \tag{4.34}$$

It is tempting to interpret this as a $\mathbb{Z}_2$ anomaly when $N$ is odd and as no anomaly when $N$ is even. However, we can add a continuous counterterm to the action

$$\frac{i\Theta}{8\pi^2} \int \text{Tr}(F_A \wedge F_A)\,. \tag{4.35}$$

The normalization here is such that for $Sp(N_f)$ background $A$ the coefficient $\Theta$ is $2\pi$-periodic. We let $\Theta$ be $\theta$-dependent

$$\Theta = \Theta^{(0)} + n\theta\,, \tag{4.36}$$

with integer $n$ to preserve the $2\pi$-periodicity of $\theta$ in $Sp(N_f)$ background. Then, under $\theta \to \theta + 2\pi$ the expression (4.35) is shifted by

$$2\pi in \int \frac{\text{Tr}(F_A \wedge F_A)}{8\pi^2} = 2\pi i \int \frac{nN_f}{2} \frac{\mathcal{P}(w_2(A))}{2} \mod 2\pi i\,. \tag{4.37}$$

When $N_f$ is odd, we can use these counterterms to cancel the shift of the action (4.34). The theory only has an anomaly when $N$ is odd and $N_f$ is even.

As in all the examples above, it can be written as a 5d action

$$\mathcal{A}(\theta, A) = \exp\left(\frac{2\pi i}{L} \int \frac{d\theta}{2\pi} \frac{\mathcal{P}(w_2(A))}{2}\right) \quad \text{with } L = \gcd(N-1, N_f, 2)\,. \tag{4.38}$$

As a result, the theory is invariant under $\theta \to \theta + 4\pi$ when $N$ is odd and $N_f$ is even, and in all other cases it is invariant under $\theta \to \theta + 2\pi$.

### 4.2.2 Interfaces

The anomaly constrains smooth interfaces between regions with different $\theta$. Suppose the two regions have $\theta$ and $\theta + 2\pi k$ for some integer $k$. The theory on the interface then has an ordinary 't Hooft anomaly of the zero-form global symmetry $Sp(N_f)/\mathbb{Z}_2$

$$\exp\left(2\pi i \frac{k}{L} \int \frac{\mathcal{P}(w_2)}{2}\right) \quad \text{with} \quad L = \gcd(N-1, N_f, 2)\,. \tag{4.39}$$

One possible choice for the theory on the interface is the 3d Chern-Simons-matter theory

$$Sp(N)_{k-N_f/2} + N_f \text{ fermions} \tag{4.40}$$

or its dual theory

$$\begin{aligned} Sp(k)_{-N} + N_f \text{ scalars} &\qquad k \geqslant 1\,, \\ Sp(N_f - k)_N + N_f \text{ scalars} &\qquad k < N_f\,, \end{aligned} \tag{4.41}$$

with an $Sp(N_f)$ invariant scalar potential. (See [16] for more details on the validity of these dualities.) All these theories have an $Sp(N_f)/\mathbb{Z}_2$ global symmetry with the anomaly (4.39) [15].

Further information about the theory along the interface can be found by considering the limits of large and small fermion masses.

When the fermions are heavy, the 4d theory is effectively an $Sp(N)$ pure gauge theory and there expects to be an $Sp(N)_k$ Chern-Simons theory on the interface, or another TQFT with the same anomaly [37].

When the fermions are massless, for $1 \leqslant N_f < N_{CFT}$ (with $N_{CFT}$ the lower boundary of the conformal window), the low-energy theory of the 4d theory is a sigma model based on $SU(2N_f)/Sp(N_f)$ [33]. The target space can be parametrized in two different ways:

$$SU(2N_f)/Sp(N_f) = \left\{ \Sigma = g\Omega g^T \,\middle|\, g \in SU(2N_f) \right\}, \tag{4.42}$$

with $\Omega$ the $Sp(N_f)$-invariant tensor, or

$$SU(2N_f)/Sp(N_f) = \left\{ \Sigma \in SU(2N_f) \,\Big|\, \Sigma = -\Sigma^T \ \text{and} \ \mathrm{Pf}(\Sigma) = 1 \right\}, \tag{4.43}$$

with $\mathrm{Pf}(\Sigma)$ the Pfaffian of the anti-symmetric matrix $\Sigma$. The kinetic term is the obvious $SU(2N_f)$ invariant one. Adding a small $Sp(N_f)$-preserving mass term for the fermions in (4.28) corresponds to adding a potential to the chiral Lagrangian. The potential is proportional to

$$-m \left( e^{i\theta/N_f} \mathrm{Tr}(\Sigma\Omega) + c.c. \right). \tag{4.44}$$

It has a minimum at $\Sigma = e^{-2\pi i k/N_f}\Omega$ when $\theta = 2\pi k$.

We are interested in the interfaces that interpolate between the vacuum at $\theta = 0$ and $\theta = 2\pi k$. For simplicity we restrict to the interfaces with $1 \leqslant k < N_f$. Following the similar analysis in [26], the interface configuration, up to symmetry transformations, is

$$\Sigma = \mathrm{diag}\left( \begin{pmatrix} 0 & e^{i\alpha_1} \\ -e^{i\alpha_1} & 0 \end{pmatrix}, \cdots, \begin{pmatrix} 0 & e^{i\alpha_{N_f}} \\ -e^{i\alpha_{N_f}} & 0 \end{pmatrix} \right). \tag{4.45}$$

The phases are divided into two groups $\alpha_1 = \cdots = \alpha_k$ and $\alpha_{k+1} = \cdots = \alpha_{N_f}$ that satisfy the constraint $\mathrm{Pf}(\Sigma) = \exp(i(\alpha_1 + \alpha_2 \cdots + \alpha_{N_f})) = 1$. The first group varies continuously from 0 to $2\pi(N_f - k)/N_f$ and the second group varies continuously from 0 to $-2\pi k/N_f$. The other configurations of the interface can be obtained by an $Sp(N_f)$ transformation $\Sigma \to g\Sigma g^T$. This shows that the theory along the interface is a sigma model based on the quaternionic Grassmannian

$$\mathrm{Gr}(k, N_f, \mathbb{H}) = \frac{Sp(N_f)}{Sp(k) \times Sp(N_f - k)}. \tag{4.46}$$

We conclude that for $1 \leqslant N_f < N_{CFT}$, the interfaces that interpolate between the vacuum at $\theta = 0$ and $\theta = 2\pi k$ with $1 \leqslant k < N_f$ has at least two phases. One is described by an $Sp(N)_k$ Chern-Simons theory and the other one is described by a nonlinear sigma model based on the quaternionic Grassmannian $\mathrm{Gr}(k, N_f, \mathbb{H})$. These two phases are captured by the theory (4.40) and its dual theory (4.41) [16].

## Acknowledgements

We thank Ofer Aharony, Thomas Dumitrescu, Jeffrey Harvey, Po-Shen Hsin, Zohar Komargodski, Gregory Moore, Kantaro Ohmori, Shu-Heng Shao, Yuji Tachikawa, Ryan Thorngren, and Edward Witten for discussions. The work of H.T.L. is supported by a Croucher Scholarship for Doctoral Study, a Centennial Fellowship from Princeton University and the Institute for Advanced Study. The work of C.C. and N.S. is supported in part by DOE grant de-sc0009988. The work of D.S.F. is supported by the National Science Foundation under Grant Number DMS-1611957. Any opinions, findings, and conclusions or recommendations expressed in this material are those of the authors and do not necessarily reflect the views of the National Science Foundation. C.C. and D.S.F. also thank the Aspen Center for Physics, which is supported by National Science Foundation grant PHY-1607611, for providing a stimulating atmosphere.

## A   Axions and Higher Group Symmetry

Throughout our analysis, we have discussed the usual presentation of anomalies via inflow. There is however another presentation of the same results by including additional higher-form gauge fields with atypical gauge transformation properties.

To carry this out for ordinary anomalies we proceed following [59]. We couple an anomalous $d$-dimensional field theory to a new $d$-form background field $A^{(d)}$ with a coupling $i \int_X A^{(d)}$. $A^{(d)}$ can be thought of as a background gauge field for a "$d-1$-form symmetry" that does not act on any dynamical field.[23] The anomaly of the $d$-dimensional theory is then formally removed by postulating that under gauge transformations of the background fields the new field transforms as $A^{(d)} \to A^{(d)} + d\lambda^{(d-1)} - 2\pi\alpha(\lambda, A)$ with $\alpha(\lambda, A)$ as in (1.1). The term with $\lambda^{(d-1)}$ is the standard gauge transformation of such a gauge field and the term with $\alpha$, which cancels (1.1), reflects a higher-group symmetry (see e.g. [59–61] and references therein).

We can apply a similar technique to our generalized anomalies involving coupling constants. Focusing on the case of the $\theta$-angle in $4d$ gauge theory, we couple our system to a classical background three-form gauge field $A^{(3)}$ through[24]

$$\frac{i}{2\pi}\theta \left( dA^{(3)} + \frac{\pi(1-N)}{N}\mathcal{P}(B) \right) . \tag{A.1}$$

Now, the lack of invariance of the original system under $\theta \to \theta + 2\pi$ is cancelled by this term. However, this term seems ill-defined. As in the general discussion above, this can be fixed by postulating that $A^{(3)}$ is not an ordinary three-form background field, but it transforms under the gauge transformation of $B$, such that the combination $F^{(4)} = dA^{(3)} + \frac{\pi(1-N)}{N}\mathcal{P}(B)$ is gauge invariant.[25] This means that the mixed anomaly between the periodicity of $\theta$ and the one-form $\mathbb{Z}_N$ global symmetry is cancelled at the cost of making the background field $B$ participate together with $A^{(3)}$ in a higher group structure [59–61].

Note that the quantum field theory does not have a conserved current that couples to this new background gauge field $A^{(3)}$. In fact, this classical background field does not couple directly to any dynamical field. Yet, such a coupling allows us to cancel the anomaly.

The use of the background three-form gauge field $A^{(3)}$ above might seem contrived. However, when $\theta$ is a dynamical field (an axion) the treatment of the anomaly involving $A^{(3)}$ is required so that there are no bulk $5d$ terms involving dynamical fields. Moreover, in this case $A^{(3)}$ is also natural from another perspective as it couples to a conserved current for a two-form global symmetry $\frac{1}{2\pi}d\theta$ [8]. Following our rule of coupling all global symmetries to background gauge fields, in this case we must introduce $A^{(3)}$.

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
