# Peer review of "Anomalies in the Space of Coupling Constants and Their Dynamical Applications II"

_SciPost Physics, doi:SciPost Phys. 8, 002 (2020)_

## Round 2 · Referee Report · Anonymous (Referee 2) · 2019-9-17

Strengths
1. Generalization of 't Hooft anomaly, which is discussed in the previous paper, is applied to 4-dimensional gauge theories.
2. Existence of 1st order phase transition while changing $\theta$ to $\theta+2\pi$ is proven. The result applies to the case with adjoint Higgs fields with $T$-breaking interactions, so this is new.
Weaknesses
Nothing
Report
This is the application of the technique developed in the previous paper with the same title. There, generalization of anomaly matching is proposed by promoting couplings as background fields. In this paper, it is applied to the physics of theta angle in 4d gauge theories, and they revisit Dashen phenomenon in a more broad setup.
About two years ago, Dashen phenomenon is clarified as a consequence of mixed anomaly (or global inconsistency) between 1-form symmetry and CP symmetry at theta=pi by Gaiotto, Kapustin, Komargodski, Seiberg (GKKS). In this derivation, CP symmetry at theta=pi is very crucial, so it cannot be applied if we add adjoint Higgs field with CP-breaking interactions. In other words, possible origin of explicit CP breaking should be due to theta angle, and any other origin was disallowed.
In this paper, although the consequence is slightly weakened, the authors remove this constraint by using the (generalized) mixed anomaly between 1-form symmetry and periodicity of theta angle. They basically claim that, since vacua at theta and theta+2pi are different as SPT protected by center symmetry, some nontrivial IR phenomena must happen between them.
The computational technique is almost based on the earlier study by GKKS, so it is easy to understand for readers who had read it. Moreover, they analyze the various simply-connected gauge groups. It is giving a nice generalization of the previous result and uncovers a new aspect of 4d gauge theories, so I can recommend the publication of this paper.

---

## Round 2 · Referee Report · Anonymous (Referee 1) · 2019-10-22

Strengths
1. Generalization of 't Hooft anomaly proposed in the companion paper is applied to four dimensional gauge theories including QCD
2. Using the generalized anomaly, the authors have shown existence of phase transition at some value of \theta-angle even if discrete symmetries (such as time-reversal) are not present
3. The results are extended to various gauge groups
Weaknesses
Nothing
Report
Anomaly matching is powerful technique to constrain phases and dynamics of quantum field theories. In the companion paper, the authors have extended notion of 't Hooft anomalies to involve space of coupling constants.
This paper discusses applications of the new anomalies to four dimensional gauge theories in order to constrain phases as varying theta and determine worldvolume anomalies on interfaces.
Section 2 mainly focuses on pure Yang-Mills theory with various gauge groups. The new anomalies, which are mixed anomalies between center one-form symmetry and periodicity of theta,
are used to show that there is a phase transition at some value of \theta. While the technique is new, the result itself for SU(N) case is not new in the sense that stronger results have been already obtained by the previous paper by Gaiotto-Kapustin-Komargodski-Seiberg using mixed anomalies between time reversal and the center symmetry at \theta =\pi. However, the new technique can be applied to systems without time reversal symmetry (or CP). This point is demonstrated in terms of Yang-Mills theories coupled to adjoint Higgs fields and therefore the results in sec.2.2 are really new.
Section 3 provides geometric perspectives of the new anomalies.
While this section requires familiarity with related mathematics,
I think that this section is useful for experts working on this topic and mathematicians.
Section 4 focuses on QCD with N_f flavors (2N_f for Sp(N) case) with equal masses. For the SU(N) case, the authors argue that the new anomaly is present if and only if the greatest common divisor between N and N_f is nontrivial. They also discuss the Sp(N) case where we have the new anomaly only for odd N and even Nf.
I expect that the technique developed here can be applied to many other systems and are therefore very important.
Thus I strongly recommend this paper for the publication in SciPost.
Requested changes
I guess that the following minor change may be useful for readers. In the current version, there is no guide to appendix in the main text. The guide could be written in e.g. the last of sec.1.4.
Typos:
1. I guess that F in eq.(2.1) should be in small letter.
2. In the sentence above eq.(4.22), "determined" -> "determine"

---

## Editorial Decision

published